# MobileUse: A Hierarchical Reflection-Driven GUI Agent for Autonomous Mobile Operation

**Ning Li**[1][*][†], **Xiangmou Qu**[2][*], **Jiamu Zhou**[2][*], **Jun Wang**[2], **Muning Wen**[1],
**Kounianhua Du**[1], **Xingyu Lou**[2], **Qiuying Peng**[2][‡], **Jun Wang**[2][‡], **Weinan Zhang**[1][‡]
[1]Shanghai Jiao Tong University  [2]OPPO Research Institute
{pengqiuying,wangjun7}@oppo.com, wnzhang@sjtu.edu.cn

## Abstract

Recent advances in Multimodal Large Language Models (MLLMs) have enabled the development of mobile agents that can understand visual inputs and follow user instructions, unlocking new possibilities for automating complex tasks on mobile devices. However, applying these models to real-world mobile scenarios remains a significant challenge due to the long-horizon task execution, difficulty in error recovery, and the cold-start problem in unfamiliar environments. To address these challenges, we propose MobileUse, a GUI agent designed for robust and adaptive mobile task execution. To improve resilience in long-horizon tasks and dynamic environments, we introduce a hierarchical reflection architecture that enables the agent to self-monitor, detect, and recover from errors across multiple temporal scales—ranging from individual actions to overall task completion—while maintaining efficiency through a Reflection-on-Demand strategy. To tackle cold-start issues, we further introduce a proactive exploration module, which enriches the agent's understanding of the environment through self-planned exploration. Evaluations on the AndroidWorld and AndroidLab benchmarks demonstrate that MobileUse establishes new state-of-the-art performance, achieving success rates of 62.9% and 44.2%, respectively. To facilitate real-world applications, we release an out-of-the-box toolkit for automated task execution on physical mobile devices, which is available at https://github.com/MadeAgents/mobile-use.

## 1 Introduction

Recent advances in multimodal large language models (MLLMs) have significantly enhanced the capability of AI systems to understand and interact with visual environments. By jointly processing text and images, MLLMs such as GPT-4V (Achiam et al., 2023), Gemini (Team et al., 2024), Claude (Anthropic, 2024), and Qwen-VL (Bai et al., 2025) have demonstrated impressive performance on a range of vision-language tasks, including visual question answering (Hu et al., 2023, 2024), image captioning (Bianco et al., 2023; Rotstein et al., 2024), and GUI control (Müller and Žunič, 2024; Qi et al., 2025). These developments open up promising opportunities for building general-purpose agents capable of interleaved visual and textual contexts perceiving and instruction following.

Among various platforms, mobile devices present a promising option for building general-purpose intelligent agents due to their vast scope and versatility. On one hand, the wide variety of apps available on mobile platforms provides an expansive environment, making it an ideal domain for developing agents capable of addressing diverse user needs. Moreover, with only a touchable screen,

---

[*]Equal contribution.
[†]Work done during an internship at OPPO.
[‡]Corresponding authors.

user interactions on mobile devices are typically constrained to a small set of intuitive actions (e.g., click, swipe, type), making the action space relatively simple compared to desktops. On the other hand, building GUI agents specifically for mobile scenarios can address real user needs by automating repetitive or complex tasks, such as filling out forms, navigating multi-step settings, that would otherwise require significant manual effort. This is particularly valuable in contexts such as accessibility support, productivity enhancement, and digital assistance. With this in mind, we'd like to develop an effective mobile GUI agent—one that can serve as a viable solution for realizing general-purpose AI.

Several recent studies have explored the potential of building mobile agents with advanced MLLMs. A significant body of work (Wu et al., 2024; Lin et al., 2024; Xu et al., 2024b; Qin et al., 2025) focuses on constructing high-quality datasets and fine-tuning MLLMs to develop specialized domain-adapted models for GUI interaction tasks. Meanwhile, frameworks such as AppAgent-v2 (Li et al., 2024a), MobileAgent-v2 (Wang et al., 2024a), and Agent-S2 (Agashe et al., 2025) decompose mobile tasks into structured decision-making pipelines, leveraging the reasoning and compositional abilities of cutting-edge LLMs and MLLMs. Despite this progress, building robust agents for dynamic mobile tasks remains a significant challenge. **(C1) Robustness in long-horizon tasks and dynamic environments:** Many mobile tasks require long-horizon execution and are inherently complex, involving a sequence of interdependent actions. Coupled with the dynamic nature of mobile environments, where elements can frequently change (e.g., due to app updates or varying screen layouts), errors are inevitable during task execution. This necessitates that mobile agents be capable of detecting failures and recovering gracefully. **(C2) Unfamiliar scenarios:** It is common for mobile agents to encounter unfamiliar apps and interfaces, particularly in cold-start settings, which require strong exploration and on-the-fly learning capabilities to operate effectively.

To address these challenges, we introduce **MobileUse**, a GUI agent designed for robust execution and flexible error recovery in dynamic mobile operation tasks. 1) We equip MobileUse with a **hierarchical reflection** architecture, which enables the agent to self-monitor, verify, and revise its decisions during the execution. Hierarchical reflection works from the microscopic step-level to the macroscopic task-level: It checks whether each action is in line with expectations, whether the process follows the correct track, and whether the goal is successfully accomplished. Besides, recognizing that excessive reflection on each step could increase the overhead and even be detrimental, we incorporate a Reflection-on-Demand strategy. By selectively invoking reflection based on a confidence score, hierarchical reflection achieves the unity of performance and efficiency. 2) Observing that many failures are caused by unfamiliarity with the environment, we design a **proactive exploration** module, enabling the agent to proactively interact with the environment prior to the task execution. It collects general knowledge on unfamiliar apps, helps the agent complete downstream tasks more efficiently and accurately in cold-start scenarios. As a whole, we build MobileUse as a multi-agent framework, including an Operator for task execution, a Progressor for progress summarization, Hierarchical Reflectors for thoughtful reflection, and a Proactive Explorer for knowledge accumulation.

We evaluate the performance of MobileUse on two dynamic Android benchmarks: AndroidWorld and AndroidLab. Empirical results show that MobileUse achieves state-of-the-art (SOTA) performance, with success rates of 62.9% and 44.2%, respectively. With comprehensive ablation studies and analyses, we highlight the effectiveness of hierarchical reflection and proactive exploration in solving complex tasks. To facilitate real-world applications, we release an out-of-the-box toolkit for automated task execution on physical mobile devices. With detailed documentation, users can easily connect their mobile phones to MobileUse and experience its capabilities as a powerful mobile assistant.

In summary, we propose **MobileUse**, a GUI agent with the following contributions:

- **Hierarchical Reflection.** We propose hierarchical reflection, a novel architecture to support robust execution and flexible error handling in mobile operation tasks. The hierarchical reflection architecture enables the agent to detect and recover from errors only when necessary, operating across multiple levels—from individual action execution to overall task completion.

- **Proactive Exploration.** We design a proactive exploration module to improve robustness against unfamiliar environments. Rather than relying on explicit instructions, our method depends on the model's proactive exploration to accumulate common knowledge for better task execution.

- **SOTA Performance and Support for Real-world Application.** We conduct comprehensive evaluations on two mobile benchmarks and demonstrate that MobileUse achieves state-of-the-art performance compared to strong baselines. To facilitate real-world applications, we release an out-of-the-box toolkit for automated task execution on physical mobile devices.

## 2 Related Works

### 2.1 GUI Agent

Autonomous agents have demonstrated significant potential in enhancing human tasks effectively. In digital environments, information involves multimodal information with text, images, and visual elements. This complexity poses challenges for language models, driving increased research interest in GUI Agents.

Models like GPT-4o (Hurst et al., 2024), Qwen2.5-VL (Bai et al., 2025), and UI-TARS (Qin et al., 2025) integrate visual understanding capabilities, enabling end-to-end execution of GUI automation tasks through natural language instructions. These models advance fine-grained visual perception, document parsing, object localization, and reasoning, laying the foundation for versatile GUI Agents. Similarly, AppVLM (Papoudakis et al., 2025), FerretUI-2 (Li et al., 2024b), and Aria-UI (Yang et al., 2024b) offer lightweight visual models, while a scalable data pipeline and pretraining on GUI tasks further enhance grounding and interaction ability. To better identify the interactive UI elements, V-Droid (Dai et al., 2025) parses the XML representation of the UI state. It utilizes an agent as a verifier rather than a conventional generator to execute appropriate actions. Advancements in multi-modal understanding, reasoning, and task automation have significantly enhanced the capabilities of GUI Agents (Anthropic, 2024), positioning them as transformative tools in human-mobile interaction.

### 2.2 Multi-Agent System

As research on automatic agents progresses, multi-agent systems are increasingly emphasized due to the inherent limitations of monolithic approaches in handling long-context scenarios involving multi-modal data. Single agents often struggle to meet the demands of planning, reasoning, and grounding tasks simultaneously. Magentic-One(Fourney et al., 2024), AutoGen(Wu et al., 2023), and CAMEL (Li et al., 2023) facilitate an orchestrator-agent framework, where the orchestrator plans and interprets user intent and generates instructions, while the assistant agent executes tool invocations. Mobile-Agent-V2 (Wang et al., 2024a) introduces a reflection agent to evaluate operator performance and suggest corrective actions when deviations occur. AppAgent (Zhang et al., 2025), Agent-S2 (Agashe et al., 2024) enhances task decomposition by a self-evolution module, leveraging web knowledge and episodic memory for long-horizon tasks. These advancements highlight the growing importance of collaborative multi-agent architectures in complex task automation.

However, the current comprehension of the reflection module is still inadequate. Reflection at the action level (Wang et al., 2024a; Wu et al., 2025) demonstrates limited utility in facilitating long-horizon tasks, while an excessive reliance on simplistic reflections not only prolongs processing time but also frequently leads to operational errors caused by hallucination-induced assessments. In addition, existing agent frameworks exploring self-evolution modules typically extract task-related experiences from previously executed trajectories. Nevertheless, these experiences often fail to be effectively transferred to the execution of new tasks, limiting their adaptability and generalization.

## 3 MobileUse

### 3.1 Framework Overview

As illustrated in Figure 1, we build MobileUse as a multi-agent framework designed for robust execution and flexible error recovery in dynamic mobile operation tasks. It consists of two stages: **Proactive Exploration stage** enables the agent to familiarize itself with the new environment while systematically exploring and accumulating common knowledge. **Autonomous Mobile Operation stage** incorporates the Operator, Hierarchical Reflectors, and the Progressor to execute user instructions in a collaborative and feedback-driven loop.

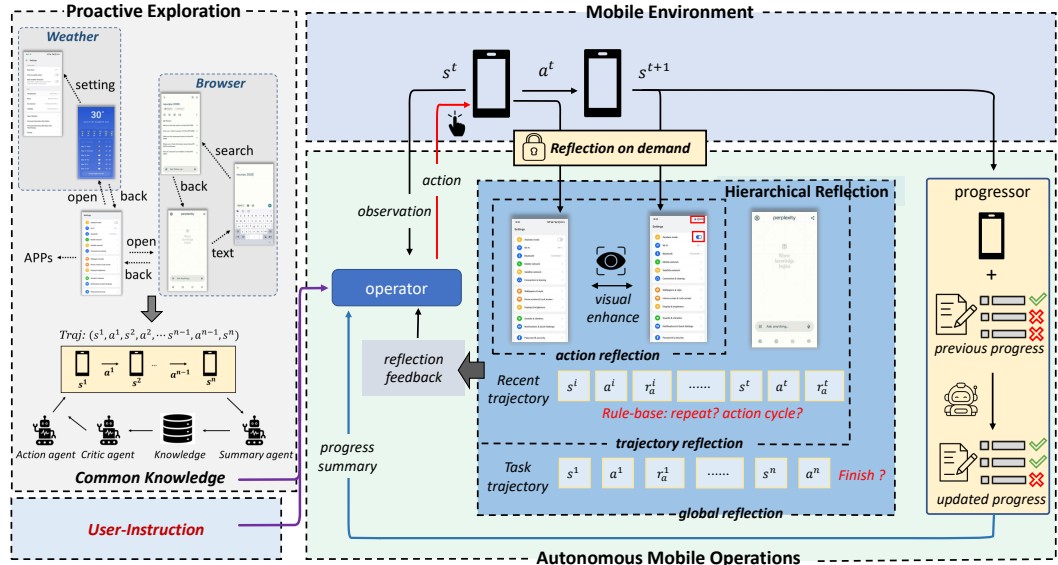

Figure 1: Overview of the MobileUse agent. In the Proactive Exploration stage, MobileUse familiarizes itself with the new environment while systematically exploring and accumulating common knowledge. In the Autonomous Mobile Operation stage, given a user instruction, in each step the Operator will observe the screenshot and output a specific action. Then MobileUse will perform hierarchical reflection at three different levels for robust task execution. Finally, the Progressor will summarize and update the current progress at the end of each step iteration.

Table 1: Notations and corresponding descriptions.

| Notation | Description |
|---|---|
| $I$ | User instruction. |
| $K$ | Knowledge collected through proactive exploration. |
| $K_I$ | Knowledge related to the user instruction $I$. |
| $E, O, R, P$ | The Explorer, Operator, Reflector and Progressor. |
| $s^t$ | Screenshot at step $t$. The superscript $t$ means the $t$-th step. |
| $a^t$ | The output generated by the Operator at step $t$. |
| $a_t^t, a_a^t, a_d^t$ | The thought, structure and natural language representation of an action. $a^t = (a_t^t, a_a^t, a_d^t)$ |
| $a_{\text{type}}^t, a_{\text{params}}^t$ | The action type and parameters. $a_a^t = (a_{\text{type}}^t, a_{\text{params}}^t)$ |
| $r_a^t, r_t^t, r_g^t$ | The feedback generated by the Action, Trajectory, and Global Reflector at step $t$. |
| $r^t$ | The feedback of hierarchical reflection at step $t$. $r^t = (r_a^t, r_t^t, r_g^t)$. |
| $p^t$ | The summarized progress generated by the Progressor at step $t$. |

Stage I: **Proactive Exploration**. In this stage, the agent is guided to proactively interact with the apps and generate common knowledge $K$ based on the execution trajectory. We do not provide user instructions for the agent to complete specific tasks, allowing the agent to fully explore the environment without being constrained by specific tasks. More details are provided in Section 3.3.

Stage II: **Autonomous Mobile Operation**. Given a user instruction $I$, firstly relevant knowledge $K_I$ will be retrieved to support downstream execution. In the $t$-th step of the task execution, the Operator $O$ serves as the decision-making agent to generate actions $a^t$ with $I$, $K_I$, current screenshot $s^t$, history actions $a^{:t-1}$, the feedback $r^{t-1}$ and progress $p^{t-1}$ from last step. Formally,

$$a^t = (a_t^t, a_a^t, a_d^t) = O(I, K_I, s^t, a^{:t-1}, r^{t-1}, p^{t-1}), \tag{1}$$

where $a_t^t$ is a thought containing the reasoning process, $a_a^t$ and $a_d^t$ are the structured and natural language descriptions of the action. See Appendix B for the full action space. After action execution, MobileUse will perform hierarchical reflection to generate reflection $r^t$ in three different levels (Section 3.2). Finally, the Progressor $P$ maintains a progress summary $p^t$ of the task execution,

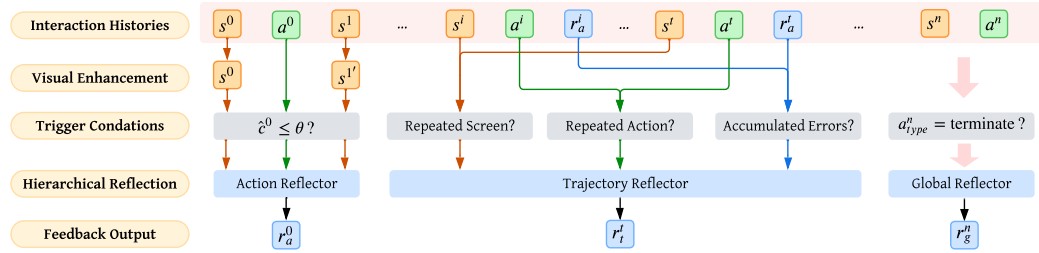

Figure 2: Overview of the hierarchical reflection architecture. The Action Reflector operates on a single step to provide immediate feedback. The Trajectory Reflector operates on the latest trajectory to ensure effective progress. The Global Reflector operates on the overall interaction history to validate the task completion.

updating it after each step by integrating past progress, the latest action, and reflection feedback:

$$p^t = P(a^{:t-1}, p^{t-1}, a^t, r^t). \tag{2}$$

This two-stage execution with the collaboration of different modules enables MobileUse to accumulate knowledge, interpret complex instructions, and adaptively self-correct, resulting in robust and reliable task execution.

## 3.2 Hierarchical Reflection

During the execution of complex mobile tasks, agents frequently encounter challenges such as incorrect grounding (Gou et al., 2025), hallucinations (Liu et al., 2023), flawed planning (Agashe et al., 2025), and getting stuck in repetitive behaviors (Agashe et al., 2024). These errors often go undetected in purely feedforward architectures and can accumulate over time, ultimately leading to task failure. To mitigate these issues, we propose a hierarchical reflection architecture that enables the agent to self-monitor, verify, and revise its decisions across multiple temporal scales. This mechanism comprises three components: Action Reflector for immediate feedback, Trajectory Reflector for progress reflection, and Global Reflector for overall validation, as illustrated in Figure 2.

### 3.2.1 Action Reflector

When operating the device, a mobile agent can make various mistakes, such as opening the wrong app, forgetting to activate the text box before typing text, clicking the wrong icon, etc. When errors occur, the screen often does not change as expected. The Action Reflector $R_{\text{action}}$ is designed to provide immediate feedback after each action execution. At each step $t$, the Action Reflector will observe the screenshots before and after the action execution to determine whether the action achieved its intended effect. To enhance perception and facilitate error localization, we design a Perception module $Pe$ to pre-compute the visual difference between the two screenshots and highlight the changed regions using bounding boxes. The Action Reflection process can be formulated as:

$$r_a^t = R_{\text{action}}(I, s^t, Pe(s^t, s^{t+1}), a^t), \tag{3}$$

where $r_a^t$ is the generated feedback that describes the error and possible causes.

In the actual task execution process, we observe that most of the actions made by the Operator are correct. Therefore, invoking the Action Reflector at every step can introduce latency and unnecessary computation. Additionally, the Action Reflector can make mistakes, which will mislead the Operator to a wrong state or repeated error correction. To address this, we propose a **Reflection-on-Demand** strategy, which dynamically determines when to trigger reflection based on the action confidence.

Specifically, in step $t$, the Operator outputs a predicted action type $a_{\text{type}}^t$, represented as a token sequence $(w_1, w_2, \ldots, w_k)$, where usually $k \in \{1, 2\}$. Each token $w_i$ is associated with a model-assigned probability $p(w_i)$. We compute the confidence score $\hat{c}^t$ for the action as the mean log-probability of its tokens:

$$\hat{c}^t = \frac{1}{k} \sum_{i=1}^{k} \log p(w_i). \tag{4}$$

Only when $\hat{c}^t \leq \theta$, where $\theta$ is a predefined threshold, the Action Reflector is invoked to reflect on the current step.

Equipped with the Action Reflector, MobileUse is capable of detecting and recovering from errors caused by one-step failure, such as failed grounding, visual hallucinations, and incorrect UI understanding.

### 3.2.2 Trajectory Reflector

Only the Action Reflector is not enough for the robust and consistent execution of complex and long-horizon mobile tasks. Sometimes, the mobile agent executes each action correctly, but it is not on the right path to finish the user instruction. In other circumstances, although the Action Reflector recognizes the error in the current step and provides some suggestions, the agent is not guaranteed to fully recover from the potential cumulated errors, resulting in a repeated failure in the near future.

Here we design a Trajectory Reflector $R_{\text{trajectory}}$, which operates over the latest progress, and a short history of recent actions (e.g., last 3–5 steps) along with the corresponding action-level reflections to evaluate whether the trajectory is coherent and progressing toward task completion. It is designed to detect error patterns that span multiple steps, such as deviations from the user instruction due to accumulated drift or repeated actions. Formally,

$$r_t^t = R_{\text{trajectory}}(I, p^{t-1}, (a^i, r_a^i), \ldots, (a^t, r_a^t)), \tag{5}$$

where $r_t^t$ is the generated feedback. $p^{t-1}$ is the progress summarized by the Progressor in the last step. $r_a$ will be empty if the Action Reflector is not invoked in the corresponding step.

The Trajectory Reflector is also called only when necessary to balance the efficiency and performance. Instead of computing a confidence score, we define several trigger conditions for the Trajectory Reflector that focus on task progress and error patterns: **(1) Repeated Actions**, indicating that the agent is stuck or making ineffective decisions. **(2) Repeated Screenshots**, suggesting that the agent is failing to make progress. **(3) Accumulated Action-Level Errors**, requiring broader analysis to assess whether the trajectory is deviating from the task goal. Once any of these conditions are detected, the Trajectory Reflector is invoked to analyze recent actions, helping the Operator stay on track and make effective progress toward fulfilling the user instruction.

### 3.2.3 Global Reflector

When the Operator believes that the user instruction has been completed, it will terminate the task execution with a specially defined action `terminate`. However, the task may not actually be finished. In a long-horizon task that consists of several sub-goals, the agent may omit some important steps. In other cases, the Operator will terminate the task prematurely due to hallucination or high complexity of the task.

The Global Reflector $R_{\text{global}}$ is designed to review the completion of tasks from a global perspective. Invoked only upon task completion, the Global Reflector evaluates whether the user instruction has been successfully fulfilled based on the historical actions and the latest screenshots. If it determines that the task remains incomplete, it will provide feedback to the Operator, who will be asked to continue finishing the task in the next iteration. Otherwise, the task will be terminated gracefully. Formally,

$$r_g^t = R_{\text{global}}(I, (a^0, r_a^0, r_t^0), \ldots, (a^t, r_a^t, r_t^t), s^j, \ldots, s^t) \quad \text{if } a_{\text{type}}^t = \texttt{terminate}, \tag{6}$$

where $r_a^i$ and $r_t^i$ will be empty if the corresponding reflector is not invoked in the $i$-th iteration. In MobileUse, we choose $j = \max(0, t-3)$ to balance the efficiency.

Together, these three-level reflection mechanisms form a hierarchical reflection architecture that allows the agent to detect and correct errors at different scales. Each reflector provides timely feedback to the Operator, enabling it to revise strategies, avoid redundant behavior, and maintain alignment with the user instruction throughout the task.

### 3.3 Proactive Exploration

When interacting with mobile devices, the agent often makes mistakes due to unfamiliarity with the environment, such as not knowing the meaning of a certain icon's color, as shown in Figure

11. A possible solution is to adopt the self-evolution mechanism (Agashe et al., 2024; Wang et al., 2025; Liu et al., 2025b), which gradually accumulates experience when completing tasks. However, erroneous experiences can interfere with the model's normal execution of actions. On the other hand, the lack of common knowledge about interactions makes it difficult for the model to correct complex operations through self-evolution.

Proactive exploration is designed to enable the agent to mimic human exploration of new environments and discover the operational common knowledge of the environment. As illustrated in the left of Figure 1, our method adopts a multi-agent collaboration framework to guide the exploration process: (i) For each app, we guide the Action Agent explores unfamiliar regions of the current app environment without relying on task-specific instructions. (ii) After a trajectory with $n$ steps $T_i = (s^1, a^1, \ldots, a^n, s^n)$ is collected by the Action Agent, the Summary Agent identifies valuable operation trajectories and summarizes them into reusable experiences $K_i = \text{summary}(s^t, a^t, \ldots, a^{t+i}, s^{t+i})$, thereby establishing an automated pipeline for data exploration and knowledge generation. We constrain these summarized experiences to be as general and beneficial as possible. (iii) The Judge Agent monitors the exploration process to prevent redundant or ineffective behavior and provides corrective feedback when necessary.

Unlike the previously proposed task-driven exploration approach (Zhang et al., 2025), which relies on manually constructed tasks, our proactive exploration is a task-agnostic and mostly autonomous method. By combining different agents, it can proactively and effectively collect general knowledge about UI layouts, navigation patterns, and potential states, which is especially useful when dealing with scenarios involving exceed the boundaries of their pre-trained capabilities.

# 4 Experiments

## 4.1 Experimental Settings

**Benchmarks.** We evaluate the performance of MobileUse on two widely-used mobile benchmarks: **AndroidWorld** (Rawles et al., 2025) and **AndroidLab** (Xu et al., 2024a). AndroidWorld includes 116 tasks across 20 apps, with randomized parameters that yield millions of possible task variants, emphasizing generalization to diverse instructions and UI states. AndroidLab covers 138 tasks from 9 apps with fine-grained success metrics and a structured evaluation pipeline. Both benchmarks provide controllable Android interaction environments, standardized task initialization procedures, and well-defined automated evaluation processes, ensuring consistency in evaluation.

**Baselines.** We compare our method against a diverse set of recent baselines, grouped into two main categories. Single-agent models are end-to-end systems that directly map textual or visual inputs to GUI operations, including general-purpose models such as GPT-4o (Achiam et al., 2023), Claude (Anthropic, 2024), Gemini (Wang et al., 2024b), and Qwen2.5-VL (Bai et al., 2025). It also includes domain-specialized models such as InfiGUIAgent (Liu et al., 2025a), Aguvis (Xu et al., 2024b), V-Droid (Dai et al., 2025), and UI-TARS (Qin et al., 2025), which are specifically trained or fine-tuned on various GUI operation datasets. Multi-agent frameworks like UGround (Gou et al., 2025), Aguvis (Xu et al., 2024b), Aria-UI (Yang et al., 2024b), and Agent-S2 (Agashe et al., 2025) decompose the task into sub-components, usually combining powerful language models (e.g., GPT-4o) with perception or grounding modules for improved decision-making.

**Implementation Details.** We use the open-source multimodal language model Qwen2.5-VL-72B-Instruct (Bai et al., 2025) with temperature $= 0$ for our base model. To assess the impact of model size, we also evaluate the performance with the Qwen2.5-VL-7B-Instruct and 32B model in Appendix D.2. Besides, we release the MobileUse Toolkit with open-source code and comprehensive documentation, which is detailed in Appendix E.

## 4.2 Main Results

Table 2 and Table 3 demonstrate that MobileUse outperforms existing mobile agents on the AndroidWorld and AndroidLab benchmarks, achieving 62.9% and 44.2% success rates, respectively. In AndroidWorld, MobileUse achieves a 3.4% performance improvement compared to the SOTA solution V-Droid. Compared to the best Multi-Agent baseline Agent-S2, which is built on a powerful close VLLM Claude-3.7-Sonnet and a carefully tuned vision model UI-TARS-72B-DPO, MobileUse achieves a 8.6% performance improvement. Compared to the single agent Qwen2.5-VL-72B-Instruct,

MobileUse significantly increases the success rate by 27.9%. In AndroidLab, MobileUse achieves a 5.9% performance improvement on the task success rate compared to the strongest baseline V-Droid. It also achieves the best result on the sub-goal success rate. The RRR and ROR are secondary metrics reflecting each agent step's redundancy and rationality. Although MobileUse is not optimal in these two aspects, it just illustrates that our hierarchical reflection mechanism can reflect and correct errors with more execution steps, achieving a better performance on the overall success rate.

Table 2: Success Rate (%) on the AndroidWorld benchmark.

|  | Method | Model | SR↑ |
|---|---|---|---|
| Single Agent | InfiGUIAgent (Liu et al., 2025a) | InfiGUIAgent-2B | 9.0 |
| | CogAgent (Hong et al., 2024b) | CogAgent-9B-20241220 | 9.0 |
| | Gemini (Wang et al., 2024b) | Gemini-1.5-Pro | 22.8 |
| | Aguvis (Xu et al., 2024b) | Aguvis-72B | 26.1 |
| | Claude (Anthropic, 2024) | Claude Computer-Use | 27.9 |
| | GPT-4o (Hurst et al., 2024) | GPT-4o | 34.5 |
| | Qwen2.5-VL (Bai et al., 2025) | Qwen2.5-VL-72B-Instruct | 35.0 |
| | UI-TARS (Qin et al., 2025) | UI-TARS-72B-SFT | 46.6 |
| | V-Droid (Dai et al., 2025) | V-Droid | 59.5 |
| Multi-Agent | M3A (Rawles et al., 2025) | GPT-4 Turbo | 30.6 |
| | UGround (Gou et al., 2025) | GPT-4o + UGround | 32.8 |
| | Aguvis (Xu et al., 2024b) | GPT-4o + Aguvis-7B | 37.1 |
| | Mobile-Agent-v2 (Wang et al., 2024a) | Qwen2.5-VL-72B-Instruct | 37.1 |
| | Aria-UI (Yang et al., 2024b) | GPT-4o + Aria-UI | 44.8 |
| | AndroidGen (Lai et al., 2025) | GPT-4o | 46.8 |
| | Agent-S2 (Agashe et al., 2025) | Claude-3.7-Sonnet + UI-TARS-72B-DPO | 54.3 |
| | MobileUse (Ours) | Qwen2.5-VL-72B-Instruct | **62.9** |

Table 3: Results on the AndroidLab benchmark. SR, Sub-SR, RRR, and ROR represent Success Rate, Sub-Goal Success Rate, Reversed Redundancy Ratio, and Reasonable Operation Ratio, respectively.

| Model | SR↑ | Sub-SR↑ | RRR↑ | ROR↑ |
|---|---|---|---|---|
| CogVLM2-ft (Hong et al., 2024a) | 11.59 | 16.06 | 57.37 | 85.58 |
| Qwen2.5-VL-72B-Instruct (Bai et al., 2025) | 17.52 | 24.00 | 73.60 | 81.92 |
| Gemini-1.5-Pro (Wang et al., 2024b) | 18.84 | 22.40 | 57.72 | 83.99 |
| Qwen2-7B-ft (Yang et al., 2024a) | 19.57 | 24.40 | 77.31 | **92.48** |
| LLaMA3.1-8B-ft (Grattafiori et al., 2024) | 23.91 | 30.31 | 75.58 | 92.46 |
| Claude-3.5-Sonnet (Anthropic, 2024) | 28.99 | 32.66 | **113.41** | 81.16 |
| GPT-4o (Hurst et al., 2024) | 31.16 | 35.02 | 87.32 | 85.36 |
| AutoGLM[4] (Liu et al., 2024) | 36.20 | - | - | - |
| V-Droid (Dai et al., 2025) | 38.30 | - | - | - |
| MobileUse (Ours) | **44.20** | **50.01** | 74.40 | 88.50 |

## 4.3 Ablation Study

Table 4: Ablation study on the AndroidWorld benchmark.

| Method | Easy Tasks | Medium Tasks | Hard Tasks | Average SR |
|---|---|---|---|---|
| Base (Operator + Progressor) | 65.6 | 41.1 | 13.7 | 49.5 |
| +) Action Reflector | 71.2 | 45.1 | 22.7 | 55.17 |
| +) Trajectory Reflector | 71.5 | 46.7 | 24.7 | 56.1 |
| +) Global Reflector | 70.5 | 52.8 | 31.6 | 58.6 |
| +) Reflection-on-Demand | 78.7 | 50.0 | 29.0 | 61.6 |
| +) Proactive Exploration | 83.6 | 47.2 | 26.3 | 62.9 |

To assess the impact of hierarchical reflection and proactive exploration on the agent's performance, we conduct a comprehensive ablation study on the AndroidWorld benchmark, as shown in Table

---

[4]AutoGLM and V-Droid didn't report the specific results on metrics other than the success rate.

4. In addition to the overall success rate, we also report the success rate on tasks with different difficulty levels. We can see that each reflector in hierarchical reflection architecture is proven to be effective, especially for medium- and hard-level tasks. The Reflection-on-Demand strategy leveraged by the Action Reflector can further improve the performance on easy tasks, suggesting that it can reduce useless or even erroneous reflection to improve the robustness. After adding these components, the result achieves a 12.1% SR improvement, demonstrating the effectiveness of our hierarchical reflection architecture. Additionally, incorporating proactive exploration further improves the SR by 1.3%, and 4.9% on the easy tasks. Proactive Exploration provides an opportunity to interact with the environment, providing valuable prior knowledge to complete the task. In Appendix F, we provide a detailed case study to demonstrate how each module takes effect in the actual task execution process.

## 4.4 Further Analysis

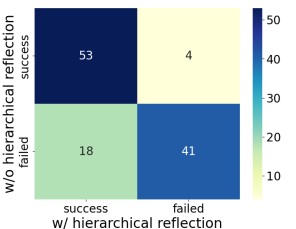

Figure 3: Confusion matrix of the task completion with and without hierarchical reflection on the AndroidWorld benchmark.

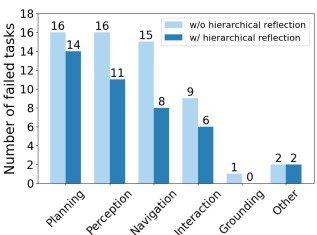

Figure 4: Error type analysis with and without hierarchical reflection on the AndroidWorld benchmark.

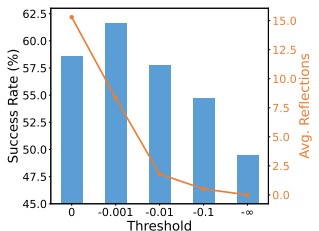

Figure 5: Performance w.r.t. different thresholds of the Reflection-on-Demand strategy on the AndroidWorld benchmark.

**Hierarchical reflection significantly enhances the agent's self-correction capability**. As depicted in Figure 3, the hierarchical reflection mechanism successfully rectified 18 failed tasks, achieving a 30.51% correction rate with minimal misjudgment at 7.02%. These findings suggest that the hierarchical reflection module not only plays a pivotal role in the effective correction of task failures but also efficiently mitigates the risk of detrimental reflections. Consequently, the efficiency gained from successful corrections significantly outweighs the inefficiencies caused by occasional misjudgments, highlighting the module's positive impact on self-correction processes.

**Hierarchical reflection reduces diverse failure modes.** In the AndroidWorld benchmark, we conducted a detailed error task analysis with and without hierarchical reflection. It can be summarized into six failure types: *Planning failures*, *Navigation failures*, *Interaction failures*, *Perception failures*, *Grounding failures*, and *Other failures*, which are detailed in Appendix C. Figure 4 illustrates the impact of hierarchical reflection on different types of task failures in the AndroidWorld benchmark. Without hierarchical reflection, failures are distributed across various categories such as planning, perception, navigation, and interaction, with relatively high counts. However, when hierarchical reflection is applied, there is a noticeable reduction in the number of failures for most error types, particularly in perception and navigation failure tasks. This indicates that hierarchical reflection effectively improves task success rates by addressing specific failure modes.

**Reflection-on-Demand strategy enhances both performance and efficiency.** Here we evaluate the effect of the Reflection-on-Demand strategy leveraged by the Action Reflector. As shown in Figure 5, the threshold $\theta = 0$ means performing Action Reflection in every step, while $\theta = \infty$ means disabling the Action Reflector. Notably, with a relatively large threshold $\theta = -0.001$, the number of reflections is reduced while the performance is further improved. This is attributed to the fact that the Reflection-on-Demand strategy can avoid some erroneous reflections when the action confidence score is high enough, as shown in Figure 10. When $\theta = -0.01$, although more than 85% of the reflections are omitted, the decrease in the success rate is less than 1.5%. These results indicate that most of the performance improvement may be attributed to some critical reflections. The Reflection-on-Demand strategy effectively identifies and preserves these key reflections, enabling great performance while significantly reducing computational overhead.

## 4.5 Efficiency

To evaluate the running efficiency of MobileUse and the impact of each module, we calculate the average execution time for each task on the AndroidWorld benchmark, as shown in Tables 5 and 6.

Table 5: Efficiency comparison of different methods on the AndroidWorld benchmark.

| Method | Time ($s$ / Task) | SR |
|---|---|---|
| M3A (screen + a11y tree) | 234 | 25.4 |
| M3A (a11y tree) | 150 | 30.6 |
| Mobile-Agent-v2 | 319 | 37.1 |
| MobileUse ($\theta = 0$) | 359 | 58.6 |
| MobileUse ($\theta = -0.01$) | 279 | 57.8 |

Table 6: Efficiency of each hierarchical reflection module on the AndroidWorld benchmark.

| Module | Time ($s$ / Task) |
|---|---|
| Action Reflector ($\theta = 0$) | 90 |
| Action Reflector ($\theta = -0.01$ ) | 10.6 |
| Trajectory Reflector | 13.2 |
| Global Reflector | 7.7 |

Here, $\theta = 0$ indicates the deactivation of the Reflection-on-Demand mechanism, meaning that the Action Reflection is performed at every step. We can find:

**MobileUse achieves both efficiency and performance gains.** Under the same backbone (Qwen2.5-VL-72B), MobileUse with Reflection-on-Demand runs faster than Mobile-Agent-v2 and achieves a much higher success rate on AndroidWorld (>20%), demonstrating the effectiveness of our hierarchical reflection design. Although M3A is relatively more efficient, this advantage is not sufficient to offset the performance gap compared to MobileUse. Furthermore, M3A includes the a11y tree as input, whereas MobileUse relies solely on screenshots, which makes it more generalizable across a wider range of mobile environments.

**Hierarchical reflection adds minimal latency.** With full reflection enabled, the Action, Trajectory, and Global Reflectors contribute approximately 25%, 3.5%, and 2% of the total time, respectively. When applying our Reflection-on-Demand strategy, the reflection overhead is significantly reduced to 10%, while the success rate drops by less than 1%, showing that unnecessary computation is effectively avoided without sacrificing performance.

**Proactive exploration achieves cost-effective efficiency.** We set the total number of exploration steps to 100 per app, with a time cost of 19.5 seconds per step, allowing each app's exploration to be completed within an hour. Notably, exploration is conducted only once per app, and the knowledge accumulated is shared across multiple downstream tasks, making the exploration cost amortizable and enhancing its practicality in real-world deployments.

## 5 Limitations and Future Work

Despite the promising results of MobileUse, there still exists some limitations: 1) MobileUse relies on the strong instruction-following, reasoning, and grounding capabilities of the foundational model, which may limit the generalizability to smaller models or edge deployment. 2) The current proactive exploration module relies on the agent's inherent capabilities to explore the environment. 3) The tasks requiring human validation (e.g., biometric login, payment confirmation) are not explicitly discussed.

In the future, we will try to improve the performance of MobileUse with smaller models by adjusting the agent framework or developing specialized vision-language models. To further optimize the proactive exploration module, we will design a reward-oriented exploration process to improve the exploration efficiency. Additionally, we will involve human-in-the-loop security confirmations into our frameworks to ensure real-world deployment.

## 6 Conclusion

In this work, we introduce MobileUse, a novel GUI agent designed to robustly automate complex mobile tasks through a hierarchical reflection architecture. Our approach addresses critical challenges in mobile environments, including long-horizon task robustness, difficulty in error recovery, and cold-start adaptation. Extensive experiments on the AndroidWorld and AndroidLab benchmarks demonstrate that MobileUse establishes new state-of-the-art performance, achieving success rates of 62.9% and 44.2%, respectively. To bridge research and real-world utility, we release the MobileUse Toolkit—a lightweight, modular system for physical mobile device automation. This enables both practical deployment and future research extensions in mobile automation.

## Acknowledgments and Disclosure of Funding

The Shanghai Jiao Tong University team is partially supported by National Natural Science Foundation of China (62322603). We also acknowledge the support of the SJTU-OPPO Joint Lab on Artificial Intelligence.

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

# A  Broader Impacts

The proposed MobileUse framework advances mobile task automation through its hierarchical reflection architecture and proactive exploration module, offering significant positive implications:

**Efficiency and Accessibility Gains**. MobileUse automates repetitive or complex tasks (e.g., form filling, multi-step navigation), reducing manual input and cognitive load. This improves efficiency in professional workflows and enhances accessibility for users with physical or visual impairments by enabling seamless interaction via automated gestures and screen parsing.

**Progress in Multimodal Interaction**. By integrating vision-language models for real-time screen understanding and task execution, MobileUse advances the state-of-the-art in mobile GUI automation. It provides a practical framework for applying multimodal models in dynamic real-world scenarios, fostering innovation in human-AI collaboration frameworks for mobile platforms.

However, current GUI agent capabilities are still limited, leading to concerns about privacy and security. The automation of mobile tasks by intelligent agents necessarily involves access to sensitive on-screen content, user inputs, and application states. Without careful design of data handling and permission control, such systems could increase the risk of privacy leakage, unauthorized actions, or exploitation by malicious actors. This highlights the importance of incorporating strong safeguards and transparent audit mechanisms.

# B  Action Space

Table 7: Action space.

| Action Type | Parameters | Description |
|---|---|---|
| key | text | Perform a key event on the device using ADB syntax. Examples include `volume_up`, `power`, `clear`. |
| click | coordinate | Click the screen at the specified `(x, y)` coordinate. |
| long_press | coordinate, time | Press and hold on the screen at `(x, y)` for a specified number of seconds. |
| swipe | coordinate1, coordinate2 | Swipe from the starting coordinate `(x1, y1)` to the end coordinate `(x2, y2)`. |
| type | text | Input text into the currently focused input box. |
| clear_text | \ | Clear the content of the active input box. |
| system_button | button | Press a system button: `Back`, `Home`, `Menu`, or `Enter`. |
| open | text | Launch an app on the device by name. |
| wait | time | Wait for a specified number of seconds. |
| take_note | text | Save observed text on the current screen for future use. |
| answer | text | Answer the user query. |
| terminate | status | Terminate the current task and report whether it was a `success` or `failure`. |

# C  Failure Types

The failure types on AndroidWorld benchmark with and without hierarchical reflection.

- *Planning failures*, whether the agent produces action is incorrect, insufficient, or early termination.
- *Navigation failures*, where the agent struggles to find a certain element or function, suggesting deficiencies in layout understanding and navigation.
- *Interaction failures*, where the agent is unable to successfully manipulate an element, reflecting a lack of domain knowledge of GUI interactions.

- *Perception failures*, where the agent is misunderstanding the text content on the screen or the function of the icon.

- *Grounding failures*, where the agent produces inaccurate coordinates for the language description provided.

- *Other failures*, the other types of failures, for example, incorrect answers.

# D   More Experimental Details

## D.1   Experiments Compute Resources

Our experiments are training-free and utilize two types of computational resources. The first type is the multi-modal large language model service, which we deploy using vLLM(Kwon et al., 2023) on a machine running Ubuntu 20.04 with 8 CPU cores, 100 GB of memory, and 4 A100 GPUs (each with 80 GB of VRAM). The second type of resource is used for running benchmark evaluations, which require 4 CPU cores and 8 GB of memory. We note that in the absence of GPU resources, the multi-modal large language model service can alternatively be replaced by third-party APIs.

## D.2   Experiments on Different Model Size

Table 8: Success Rate (%) of MobileUse with different model sizes on the AndroidWorld benchmark.

| Model | Success Rate |
| --- | --- |
| MobileUse (Qwen2.5-VL-7B-Instruct) | 21.6 |
| MobileUse (Qwen2.5-VL-32B-Instruct) | 44.4 |
| MobileUse (Qwen2.5-VL-72B-Instruct) | 62.9 |

Here we change the backbone VLLM of MobileUse to assess how model size affects the agent's performance, as shown in Table 8. When using Qwen2.5-VL-7B-Instruct, the success rate becomes very low. This is because the 7B model lacks the basic instruction-following ability. The hierarchical reflection and proactive exploration mechanism are almost useless with the 7B model. The performance of MobileUse with the Qwen2.5-VL-32B-Instruct model improves a lot. Insufficient grounding and complex reasoning capabilities prevent it from achieving better performance. With the 72B model, MobileUse achieves the SOTA results. With accurate grounding and powerful reasoning ability of the foundation model, MobileUse can realize its huge potential for solving complex mobile operation tasks.

# E   MobileUse Toolkit

We develop the MobileUse Framework as a lightweight, modular, and pluggable toolkit. Based on Android Debug Bridge (ADB), this toolkit achieves seamless connectivity with a physical mobile device. Integrated with Gradio(Abid et al., 2019), it provides a visual web interface where users can input commands through a web browser to drive automated smartphone operations and monitor the execution process of the agent in real time. The MobileUse Toolkit enables end-users to achieve one-click smartphone automation and intuitively experience the capabilities of GUI Agents. Furthermore, its flexible design supports the activation, deactivation, and customization of reflection mechanisms at various stages of hierarchical reflection, offering researchers an effective platform for experimentation and extension.

As shown in Figure 6, to use the Tookit, firstly users need to connect their phone to the computer via ADB. Next, install the MobileUse ToolKit on the computer and launch the WebUI service, which can be opened with the URL http://127.0.0.1:7860 in a browser. After that, configure the VLM service on the webpage, enter task commands in the input box, and click run. At this point, the MobileUse agent will automatically operate the phone and display the execution process on the webpage.

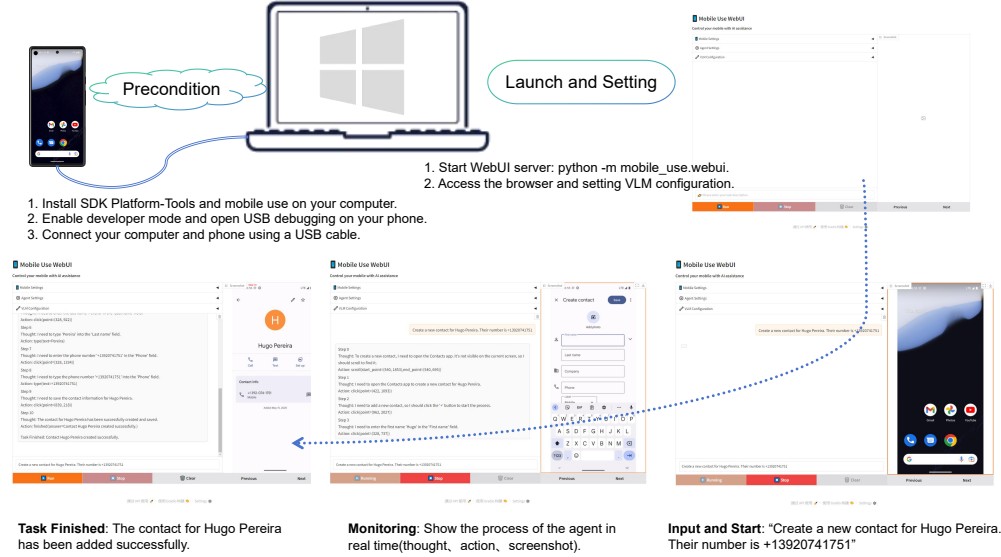

Figure 6: The illustration of MobileUse Toolkit automating mobile phone operations via ADB.

# F  Case Study

Figure 7, 8, and 9 show how the Action, Trajectory, and Global Reflector work to correct errors for robust task execution. Figure 10 shows that the Reflection-on-Demand strategy can avoid incorrect reflection. Figure 11 shows how proactive exploration can provide useful knowledge to help with the task execution.

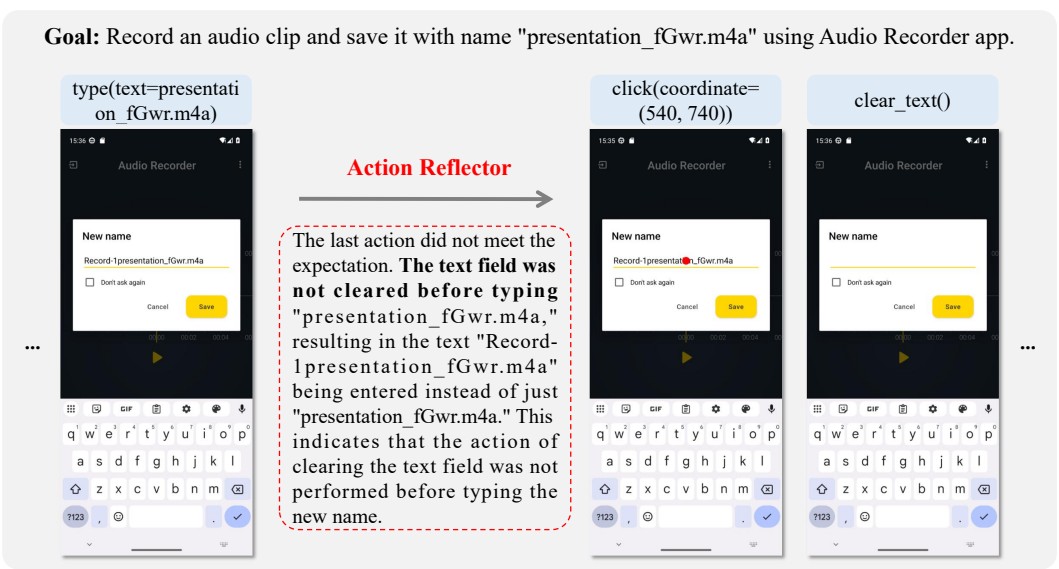

Figure 7: A case of the Action Reflection. The Operator doesn't realize that there is a default name in the input box and enters the new file name directly. The Action Reflector detects this error and provides feedback to the Operator. In the next step, the operator successfully clears the text box and enters the correct file name based on the feedback information.

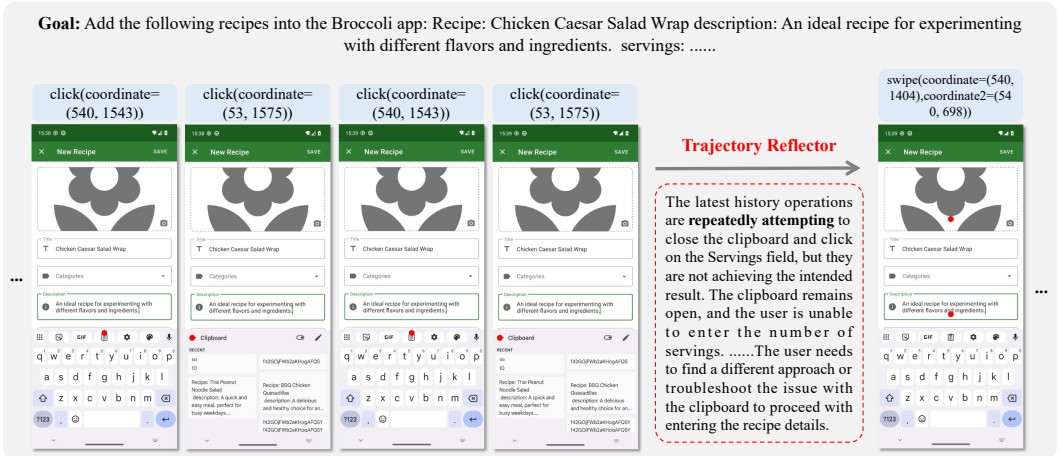

Figure 8: A case of the Trajectory Reflection. The Operator wants to click on the input box below, but due to the presence of the keyboard, the Operator is stuck in the wrong action loop. The Trajectory Reflector detects repeated actions and guides the operator in trying different operations. Finally, the Operator finds the new input box by swiping up.

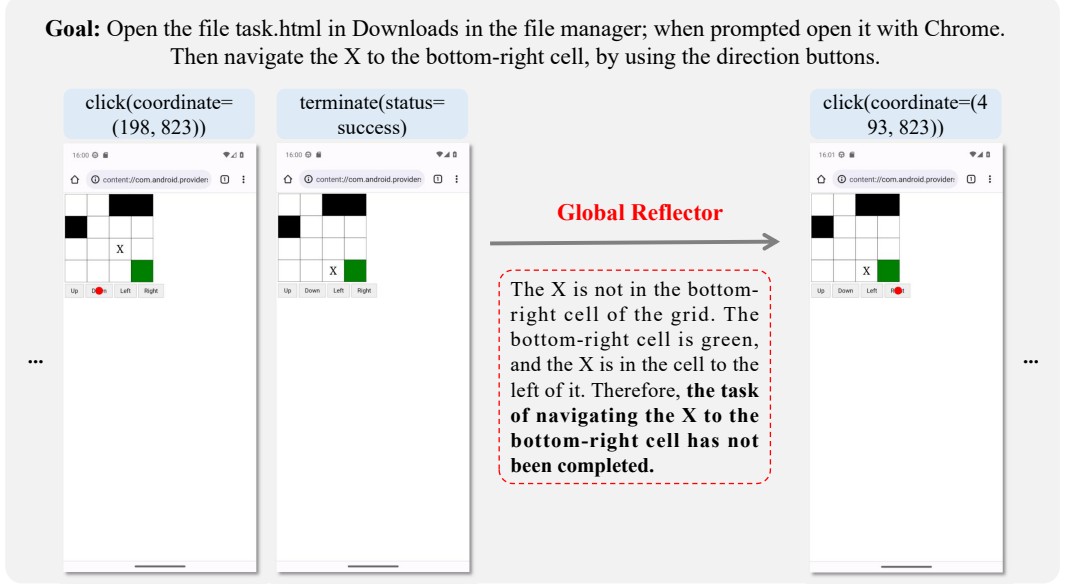

Figure 9: A case of the Global Reflection. The operator doesn't complete the user's instruction and ends the task prematurely. The Global Reflector successfully detects this error and feeds back to the operator. Finally, the task is successfully finished.

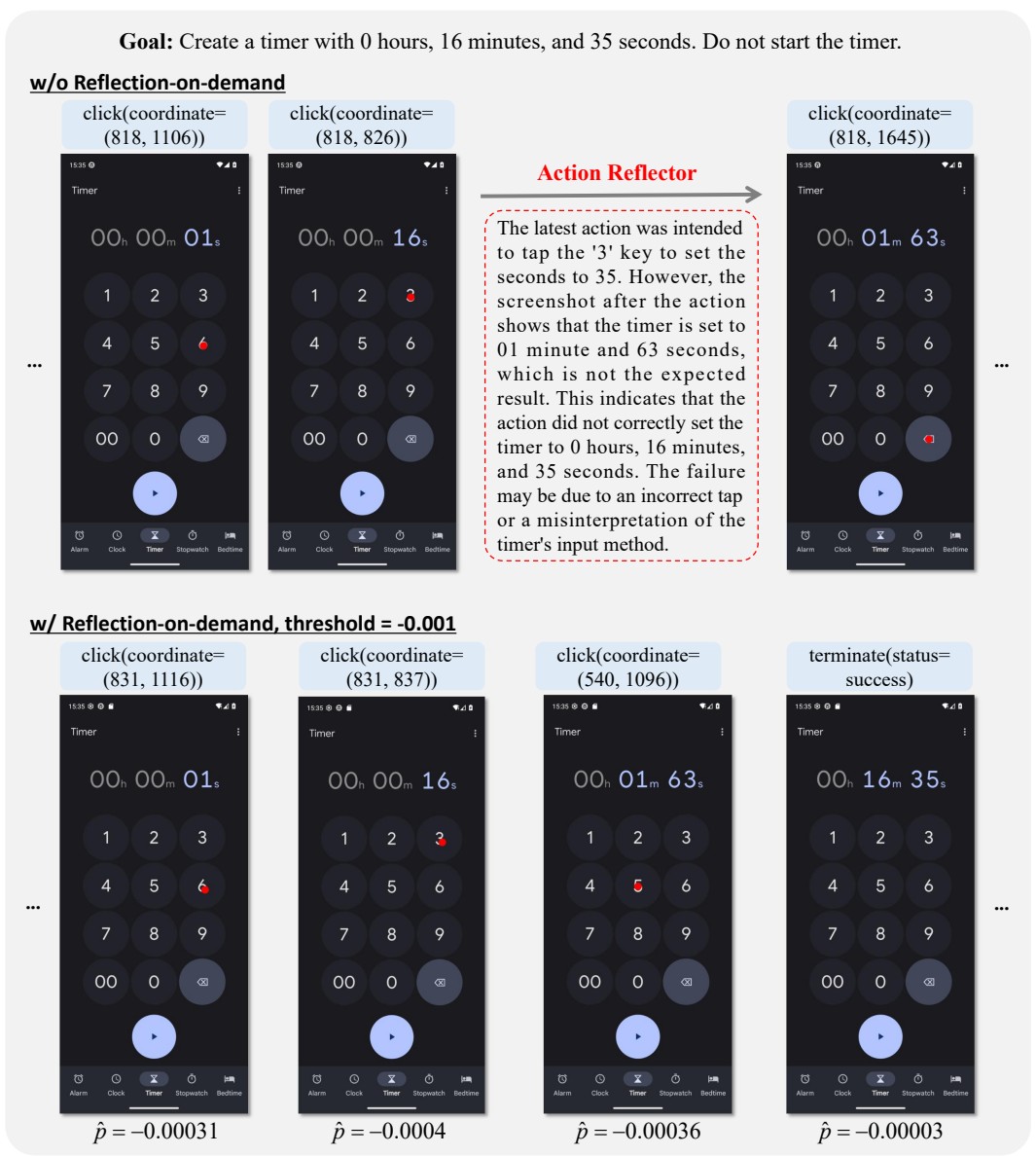

Figure 10: A case of the Reflection-on-Demand strategy. When the Reflection-on-Demand strategy is not used, the Action Reflector generates a wrong reflection, causing the Operator to make an incorrect action in the next step. When the Reflection-on-Demand strategy is used, since the confidence score of each step is higher than the threshold, the Action Reflector is not triggered, and the Operator successfully completes the task.

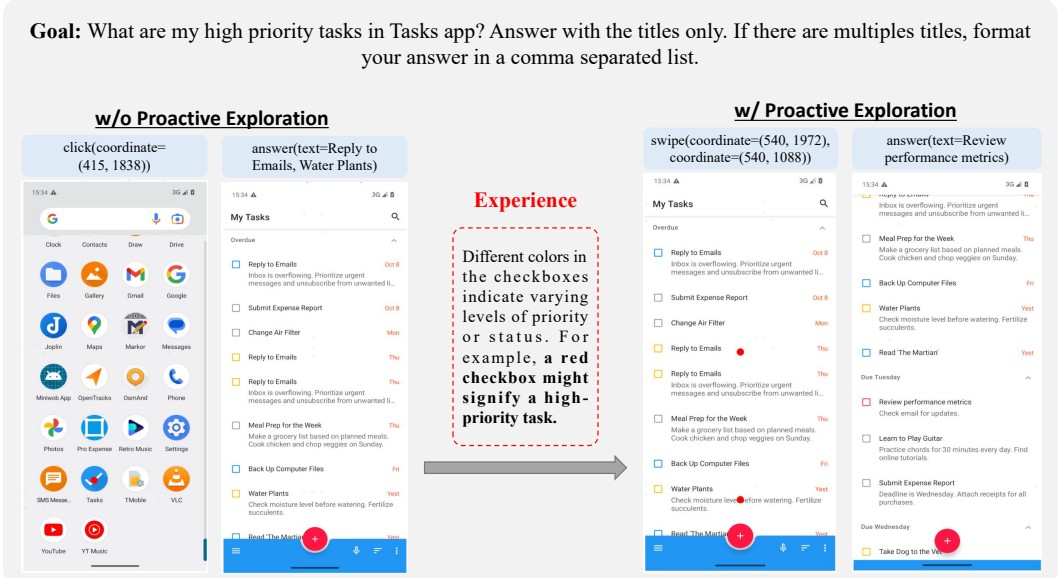

Figure 11: A case of the Proactive Exploration. Without exploration of the task app, the agent doesn't know which tasks are high priority and gives incorrect answers. After performing proactive exploration, the agent acquires knowledge about high-priority tasks, which helps him find and identify high-priority tasks during task execution and finally generates the correct answer.

# G    Examples of the Collected Knowledge in the Proactive Exploration Stage

Figure 12, 13, and 14 show the Knowledge collected from the Tasks, RetroMusic, and Camera apps.

---

**Knowledge collected from the Tasks app.**

1, Clicking the microphone icon in the App main page does not seem to produce any visible changes in the current view. The task list remains the same, and no new elements or options appear. This suggests that the microphone icon might be intended for voice input or commands, but its effects are not visually represented in the current display.

2, The App main page displays a list of tasks with checkboxes for completion status. Different colors in the checkboxes indicate varying levels of priority or status. For example, a red checkbox might signify a high-priority task, while a yellow checkbox could indicate a medium-priority task. A blue checkbox with a checkmark signifies a completed task.

3, The task list includes detailed descriptions for each task, such as "Send an update" for attending a training session and "Follow up with others" for client follow-up. These descriptions provide additional context for the tasks.

4, The bottom of the App main page features a floating action button with a plus sign, which is likely used to add new tasks. Other icons at the bottom, such as the microphone and menu icons, offer additional functionalities, though their specific effects are not visually apparent in the current view.

5, In the task creation section, clicking on the "Task name" field allows for the input of a new task name. The keyboard appears, enabling the user to type the task details. The task creation section also includes options for setting a start date, due date, and priority level, as well as adding a location and tags. The priority levels are indicated by different colored circles: gray for low, blue for medium, yellow for high, and red for critical.

6, In the location selection interface, clicking the red circular icon with a location symbol centers the map on the current location. This action does not change the visual elements significantly but adjusts the map's focus to the user's current geographical position. The interface includes a search bar for entering location names and a prompt to "Select this location," indicating the user can confirm the chosen location for further actions.

7, When a "Missing permissions" dialog appears, indicating that location permissions are needed to find the current location, clicking the "OK" button does not dismiss the dialog. This suggests that the App requires explicit permission handling for location services, and the user must grant these permissions through the device's settings or a subsequent prompt.

8, In the certificate verification interface, the App prompts the user to trust an unknown certificate. The interface displays the X509 certificate details, including the issuer, validity period, and fingerprints. The user can choose to accept or reject the certificate. Selecting the "ACCEPT" option does not change the interface, indicating that additional steps or conditions, such as manual verification of the fingerprint, might be required before the certificate is trusted.

Figure 12: Knowledge collected from the Tasks app.

**Knowledge collected from the RetroMusic app.**

1, The App main page provides options like History, Last added, Most played, and Shuffle, each with distinct icons for quick access.

2, The Suggestions section displays a New Music Mix and other music options, allowing users to explore new tracks.

3, The Recent artists section lists artists like George, enabling users to access their music directly.

4, The Recent albums section shows albums such as Music, providing easy navigation to recent music collections.

5, The bottom navigation bar includes icons for For you, Music, and other categories, facilitating seamless navigation within the App.

6, The artist detail section for George includes options like Play all and Shuffle, allowing users to listen to the artist's music in different ways. The section also displays the artist's album, providing direct access to their music collection.

7, The Song section under the artist detail for George lists individual tracks like "City of Stars," enabling users to play specific songs directly. The interface supports swiping to reveal more content, enhancing user interaction and exploration of the artist's music.

8, The artist detail section for George includes a visual representation of the artist with a placeholder icon, providing a clear and organized view of the artist's information and music options.

9, Clicking on a song title in the Song section under the artist detail for George initiates playback of the selected track, as indicated by the now-playing bar at the bottom of the App main page. This feature allows users to start listening to a specific song immediately.

10, The album detail section for Music includes options like Play all and Shuffle, allowing users to listen to the album's music in different ways. The section also lists individual songs like "Bright Lights" and "Chasing Shadows," enabling users to play specific tracks directly. The interface supports swiping to reveal more content, enhancing user interaction and exploration of the album's music.

11, The artist detail section for George includes a visual representation of the artist with a placeholder icon, providing a clear and organized view of the artist's information and music options. The interface supports swiping to reveal more content, enhancing user interaction and exploration of the artist's music.

12, The now-playing interface shows the currently playing song, "City of Stars" by George, with a progress bar indicating the elapsed time. It includes playback controls such as pause, skip, and shuffle, allowing users to manage their music playback. Additional options like adding the song to a playlist or queue are also available, enhancing the user's music listening experience.

13, The now-playing interface also features a three-dot menu that reveals additional options for the song, including Playback Speed, Drive mode, Go to album, Go to artist, Add to playlist, Save playing queue, Share, Set as ringtone, Clear playing queue, Details, Go to Lyrics, Equalizer, and Delete from device. These options provide users with extensive control over their music experience and management.

14, The Share option in the now-playing interface allows users to choose what they want to share, such as the audio file, a status update about the currently playing song, or a story. This feature enhances the user's ability to share their music experience with others.

15, After selecting the "Share story" option, the interface displays a preview of the story to be shared, including the song title "City of Stars" and the artist name "George." The "Share to Stories" button allows users to finalize and share the story, providing a seamless way to share their current music experience.

16, The interface after clicking the back arrow from the "Share story" preview returns to the now-playing interface, where users can continue managing their music playback and access additional options. This ensures a smooth transition and maintains the user's control over their music experience.

17, The now-playing interface includes a lyrics section that can be accessed by clicking the lyrics icon. If no lyrics are found, a message indicating "No lyrics found" is displayed. This feature allows users to view synchronized lyrics while listening to the song, enhancing their music listening experience.

18, Swiping on the now-playing interface reveals more content, such as the progress bar and additional playback controls, enhancing the user's ability to manage their music playback and explore more options.

19, The now-playing interface includes a sleep timer feature that allows users to set a specific duration for the music to play before stopping. Users can adjust the timer using a slider and choose whether to finish the last song before stopping. The "Start" button initiates the sleep timer, providing a convenient way to manage music playback during sleep or rest periods.

20, After setting the sleep timer and returning to the now-playing interface, users can continue managing their music playback and access additional options, ensuring a seamless and uninterrupted music experience.

21, The now-playing interface includes a "Now playing queue" section that displays the current song and upcoming tracks. Users can manage the queue by adding or removing songs, and there is a "Clear queue" button to remove all songs from the queue. This feature provides users with control over their music playlist and allows for easy queue management.

Figure 13: Knowledge collected from the RetroMusic app.

**Knowledge collected from the Camera app.**

1, The App main page provides options for selecting different modes, such as "MODE LIST" and "FILMSTRIP". Clicking "MODE LIST" reveals a menu with "Camera" and "Video" options, indicating the ability to switch between photo and video modes.

2, The App main page includes a camera icon at the bottom, which likely serves as a shortcut to capture photos or videos directly.

3, A settings icon is visible, suggesting access to configuration options for adjusting resolution, quality, and advanced features.

4, The "Z+" and "Z-" buttons at the top indicate functionality for zooming in and out, enhancing the user's control over the view.

5, The App main page displays a room scene with a TV, bookshelf, and window, providing a visual context for the camera's view.

6, After clicking "MODE LIST", the App main page transitions to a view where the camera is ready to capture images, with the camera icon at the bottom becoming active and ready for use. The three-dot menu at the bottom right likely provides additional options or settings related to the current mode.

7, Clicking "FILMSTRIP" transitions the App to a new view, which appears to be a blank or loading state, possibly indicating a section for viewing or managing captured images or videos.

8, The three-dot menu in the top right corner of the App main page reveals a "Details" option, suggesting additional information or settings can be accessed from this menu.

Figure 14: Knowledge collected from the Camera app.

