# OpenReview forum: "MobileUse: A Hierarchical Reflection-Driven GUI Agent for Autonomous Mobile Operation"
_NeurIPS.cc/2025/Conference — NeurIPS 2025 poster_

### Official Review · Reviewer_Fa8z · 2025-06-25

**Clarity:** 3
**Significance:** 2
**Originality:** 2
**Rating:** 3
**Confidence:** 4

**Summary:**

The paper proposes MobileUse, a hierarchical reflection-driven GUI agent for autonomous mobile task execution. The key contributions include a multi-level reflection mechanism for error detection and recovery, as well as a proactive exploration module to handle cold-start scenarios. Experimental results demonstrate that MobileUse achieves state-of-the-art performance on AndroidWorld and AndroidLab benchmarks. The approach is well-motivated and addresses important challenges in mobile agent robustness and adaptability.

**Questions:**

- In Eq. (3), does Pe(s^(t+1)) have only one input, s^(t+1)?

**Ethical Concerns:**

["NO or VERY MINOR ethics concerns only"]

**Final Justification:**

After the rebuttal, I still believe that this paper has issues regarding its novelty. First, as the authors themselves stated, autonomous exploration and reflection are two separate modules; however, the title of the paper only mentions reflection. I think that the extensive discussion and presentation of autonomous exploration diminish the perceived novelty of the reflection module and weaken the validity of the experimental results. Furthermore, numerous papers on reflection have already been proposed, as pointed out by Reviewer 1iG5, which further reduces the paper's originality. The authors have not clearly explained how their work differs from these existing studies. Therefore, I am inclined to give a score of 3 and do not recommend acceptance. **Finally, I kindly ask the AC to review the comments from Reviewer c8Yj. In my opinion, those comments are unreliable—being overly general and appearing to be generated by GPT.**

**Limitations:**

yes

**Quality:**

2

**Strengths And Weaknesses:**

### Strengths
- This paper addresses an important issue in the GUI Agent scenario: how to enable models to possess reflective capabilities.
- This paper conducts sufficient ablation experiments to demonstrate the necessity and effectiveness of the reflection mechanism.

### Weaknesses

I believe the main weaknesses of this paper lie in its methodology and experimental sections:

- 1. The Stage I: Proactive Exploration section lacks novelty, as the idea of exploration for document collection has already been used in prior work such as AppAgent [1]. While I agree that enabling the model to explore the environment is necessary, the proposed method seems unrelated to enhancing the model's reflective ability. If the authors aim to improve the model's reflection through exploration, I suggest referring to work [2], which provides a compelling approach and implementation for achieving this. (I do not intend to ask the authors to compare with concurrent work, but I believe this particular work offers a valuable perspective.)
- 2. Moreover, the experimental section suffers from issues common in many current works in the GUI Agent field, including unfair comparisons and a lack of implementation details. First, the evaluation protocol is unclear — it is not specified whether the evaluation was automated or manual. I recommend the authors provide more detailed implementation descriptions, as the information in the appendix is insufficient. Additionally, the range of baselines is limited; for example, End-to-End agents such as CogAgent [3] are missing. Furthermore, multi-agent frameworks like AppAgent [1] and Mobile-Agent-v2 [4] are not included in the comparison. These comparisons should also control for backbone consistency, whereas the authors present results based on Qwen-72B in the main paper, which complicates fair evaluation.

[1] Appagent: Multimodal agents as smartphone users. CHI 2025.

[2] GUI-Reflection: Empowering Multimodal GUI Models with Self-Reflection Behavior. Arixv.

[3] Cogagent: A visual language model for gui agents. CVPR 2024.

[4] Mobile-agent-v2: Mobile device operation assistant with effective navigation via multi-agent collaboration. NeurIPS 2024.

---

> ### Author Rebuttal · Authors · 2025-07-31
>
> We sincerely thank the reviewer for the constructive and detailed feedback. We appreciate the recognition of our motivation and sufficient experiments. Below, we address the main concerns point by point:
>
> ---
>
> > Weakness 1.1: The Stage I: Proactive Exploration section lacks novelty, as the idea of exploration for document collection has already been used in prior work such as AppAgent [1].
>
> We appreciate the reviewer’s concern and the opportunity to clarify this point.
>
> We'd like to clarify that our proposed proactive exploration method has **significant and fundamental differences** with prior work, such as AppAgent [1]. The exploration in AppAgent is a **task-driven approach that relies on manually constructed tasks** and thus heavily depends on human knowledge and is constrained by the quality and diversity of the predefined tasks. Whereas our proactive exploration is an **automatic and experience-oriented multi-agent collaborative module without human involvement**, which is a significant advantage in extendability and is an important quality for achieving a super intelligent GUI agent.
>
> To be specific, we highlight our proactive exploration in two aspects:
> - **Task-agnostic and fully autonomous exploration**: Proactive exploration is not given explicit tasks but instead proactively explores the environment to build general knowledge about UI layouts, navigation patterns, and possible states. In contrast, manually constructed task-oriented approaches are inherently limited by the diversity and complexity of the generated tasks.
> - **Exploration among collaborative multi-agents**: Multi-agent collaboration can more comprehensively address the demands of complex tasks, and we extend this idea to the exploration phase. i) We guide the Action Agent explores unfamiliar regions of the current app environment without relying on task-specific instructions; ii) The Summary Agent analyzes the environment states before and after actions, along with executed action information, to generate and update reusable knowledge; iii) The Judge Agent monitors the exploration process to prevent redundant or ineffective behavior and provides corrective feedback when necessary.
>
> We will include a more detailed description of the innovative aspects of our proactive exploration in the revised paper.
>
> ---
>
> > Weakness 1.2: While I agree that enabling the model to explore the environment is necessary, the proposed method seems unrelated to enhancing the model's reflective ability. If the authors aim to improve the model's reflection through exploration, I suggest referring to work [2], which provides a compelling approach and implementation for achieving this. (I do not intend to ask the authors to compare with concurrent work, but I believe this particular work offers a valuable perspective.)
>
> We thank the reviewer for this thoughtful comment.
>
> First, we would like to clarify that in our framework, the proactive exploration module and the hierarchical reflection module are designed as **two relatively independent components**. Specifically:
> - The exploration module operates before task execution. Its goal is to allow the agent to proactively explore the mobile environment without predefined tasks, so it can accumulate common knowledge about diverse UI elements, layouts, and possible interactions, which is leveraged by the Operator to be familiar with the environment.
> - The reflection module works during task execution. It uses a hierarchical reflection mechanism to continuously monitor, detect, and correct errors in the agent’s behavior in real-time. This enables the agent to recover from mistakes and adapt to unexpected changes in the environment.
>
> **Both modules target the same overarching challenge: improving the robustness and adaptability of GUI agents in real-world mobile tasks.** The exploration module reduces the chance of failure caused by unfamiliar interfaces, while the reflection module ensures the agent can self-correct and recover when errors still occur during execution. Together, they form an integrated framework — MobileUse — that addresses different aspects of the robustness problem from both a proactive (pre-task) and reactive (in-task) perspective.
>
> We also want to clarify that **the main contribution of our paper is the design and implementation of the hierarchical reflection mechanism**, which is why it takes up the majority of the paper’s technical discussion, analysis, and ablation studies. The exploration module plays a supporting role, providing additional gains by preparing the agent with broader environmental knowledge.
>
> We sincerely thank the reviewer for suggesting GUI-Reflection [2]. We will cite this paper and clarify these connections more explicitly in our revised paper.
>
> ---
>
> > Weakness 2.1: Moreover, the experimental section suffers from issues common in many current works in the GUI Agent field, including unfair comparisons and a lack of implementation details. First, the evaluation protocol is unclear — it is not specified whether the evaluation was automated or manual. I recommend the authors provide more detailed implementation descriptions, as the information in the appendix is insufficient.
>
> We thank the reviewer for highlighting the importance of a clear and fair evaluation protocol.
>
> We would like to clarify that we evaluate MobileUse on AndroidWorld [5] and AndroidLab [6], which are widely adopted standard benchmarks for GUI agent research [7,8]. Both benchmarks provide a unified testbed, including:
> - A consistent Android device environment with controlled settings;
> - Standardized task initialization procedures;
> - A well-defined automated evaluation process.
> Specifically, after the agent finishes a task or reaches the pre-defined maximum step limit, AndroidWorld and AndroidLab automatically assess the result by reading the internal system states of the Android device to determine whether the task goal has been achieved. For certain tasks in AndroidLab, the benchmark also uses GPT-4o as part of the automatic evaluation pipeline. **No manual intervention is involved in any part of the evaluation process**, which ensures that the evaluations are fully automated and consistent across different methods.
>
> To further support reproducibility and transparency, we have **open-sourced the entire benchmark setup and evaluation scripts** in our code repository under the `mobile_use/benchmark` directory. We also provide step-by-step instructions to help other researchers reproduce our results.
>
> In the revised paper, we will provide more details about the evaluation protocol. We thank the reviewer again for this constructive suggestion.
>
> ---
>
> > Weakness 2.2: Additionally, the range of baselines is limited; for example, End-to-End agents such as CogAgent [3] are missing. Furthermore, multi-agent frameworks like AppAgent [1] and Mobile-Agent-v2 [4] are not included in the comparison. These comparisons should also control for backbone consistency, whereas the authors present results based on Qwen-72B in the main paper, which complicates fair evaluation.
>
> We thank the reviewer for this valuable suggestion on expanding our baseline comparisons.
>
> We'd like to clarify that we have **included comparisons with several recent and strong End-to-End agents and multi-agent frameworks**, as shown in Table 2. For completeness, we have now added the results for CogAgent [3] and Mobile-Agent-v2 [4] on the AndroidWorld benchmark, as shown in the table below. Although CogAgent is fine-tuned for GUI automation tasks, it performs poorly on the AndroidWorld benchmark due to its insufficient grounding and reasoning ability.
>
> When comparing with Mobile-Agent-v2, we control a consistent backbone model. The result shows that our MobileUse achieves significant performance improvement compared to the Mobile-Agent-v2 framework, attributed to the hierarchical reflection architecture for error recovery and proactive exploration for common knowledge accumulation.
> |Method|Model|SR|
> |-|-|-|
> |CogAgent|CogAgent-9B-20241220|9.0|
> |Mobile-Agent-v2|Qwen2.5-VL-72B|37.1|
> |MobileUse|Qwen2.5-VL-72B|62.9|
>
> We agree that backbone consistency is an important concern. Since different methods choose various backbones and many of them are not open-sourced, we present the best result of each baseline reported in their papers to reach a fair comparison.
>
> We appreciate the reviewer’s careful attention to this issue and will incorporate these clarifications and additional results in the revised paper to make our evaluation more comprehensive and transparent.
>
> ---
>
> > Question 1: In Eq. (3), does Pe(s^(t+1)) have only one input, s^(t+1)?
>
> The Perception module takes as input both the screenshots before and after the action execution, i.e., it should be written as Pe(s^t, s^(t+1)) instead of only s^(t+1). We appreciate the reviewer catching this oversight and will correct this notation in the revised version of the paper.
>
> ---
>
> [1] Appagent: Multimodal agents as smartphone users. CHI 2025.
>
> [2] GUI-Reflection: Empowering Multimodal GUI Models with Self-Reflection Behavior. Arixv.
>
> [3] Cogagent: A visual language model for gui agents. CVPR 2024.
>
> [4] Mobile-agent-v2: Mobile device operation assistant with effective navigation via multi-agent collaboration. NeurIPS 2024.
>
> [5] AndroidWorld: A Dynamic Benchmarking Environment for Autonomous Agents. ICLR 2025.
>
> [6] AndroidLab: Training and Systematic Benchmarking of Android Autonomous Agents. Arxiv.
>
> [7] Aguvis: Unified Pure Vision Agents for Autonomous GUI Interaction. ICLR 2025.
>
> [8] AutoGLM: Autonomous Foundation Agents for GUIs. Arxiv.

---

> > ### Comment · Reviewer_Fa8z · 2025-08-06
> >
> > Thank you for the author's response. The concerns regarding the experimental section have been addressed, and I will slightly adjust my score accordingly.

---

> > > ### Author Response · Authors · 2025-08-06
> > >
> > > Thank you very much for your thoughtful feedback and for taking the time to consider our rebuttal. We genuinely appreciate your acknowledgment and the score adjustment — it is very encouraging for us.

---

### Official Review · Reviewer_1iG5 · 2025-06-30

**Clarity:** 3
**Significance:** 2
**Originality:** 3
**Rating:** 4
**Confidence:** 5

**Summary:**

This paper introduces MobileUse, a GUI agent leveraging MLLMs to automate complex tasks on mobile devices. Addressing key challenges such as long-horizon task execution, error recovery, and the cold-start problem, MobileUse incorporates a hierarchical reflection architecture for dynamic self-monitoring and efficient error handling at multiple temporal scales. It further features a proactive exploration module, allowing the agent to adapt to new environments through self-guided exploration. Evaluations on AndroidWorld and AndroidLab benchmarks show MobileUse outperforms prior methods on most metrics.

**Questions:**

This paper claims that Proactive Exploration can help accumulate common knowledge, but its experimental results are not satisfactory. Proactive Exploration also incurs a time cost. If there is a domain gap between the environment explored and the experimental environment, can Proactive Exploration still be effective?

**Ethical Concerns:**

["NO or VERY MINOR ethics concerns only"]

**Final Justification:**

Through the rebuttal, the authors have re-clarified the technical contributions of the paper, time overhead, and some module contributions. The subtle differences clarified between the proposed approach and existing methods do not constitute strong evidence of a technical contribution. Based on the experimental results provided by the authors, their method indeed introduces additional computational overhead compared to the baseline. Moreover, the authors explained that AndroidWorld has limitations on the maximum number of steps, suggesting that their method would not introduce excessive time overhead.    However, this reasoning is unreasonable because such constraints do not exist in real-world scenarios, which raises my concerns about the practicality of this method. The authors' rebuttal is detailed and addresses some issues, and the method demonstrates good accuracy on AndroidWorld, so my final score is 4.

**Limitations:**

Yes

**Quality:**

3

**Strengths And Weaknesses:**

Strengths:
1. The paper is well-written and clearly articulated, and the supplementary material provides detailed information about the experimental implementation.

2. This paper achieves strong performance on both the commonly used AndroidWorld and AndroidLab datasets.

Weaknesses:
1. The technical contribution of this paper is limited. The main contributions lie in incorporating reflectors into the GUI automation framework and leveraging historical operation information to improve operational accuracy, both of which have already been explored in prior work [1,2,3].
2. The paper is overly engineering-oriented. The method relies on multiple rounds of inference to achieve better performance at the expense of efficiency, which may negatively impact its practical applicability.
3. Introducing three reflectors in the GUI automation framework inevitably increases inference time. A quantitative comparison should be provided, but such analysis is currently missing in the paper.
4. As shown in the ablation results (Table 4), the model in the fourth row achieves better performance on Medium and Hard tasks, whereas Reflection-on-Demand and Proactive Exploration primarily benefit Easy tasks and can even significantly reduce performance on more difficult tasks. The paper lacks any explanation for this phenomenon. Since Easy tasks constitute the majority in the AndroidWorld benchmark, the full model appears strong overall, but these results suggest that Reflection-on-Demand and Proactive Exploration offer limited benefit for more challenging scenarios.
5. The paper does not sufficiently describe the implementation details of Proactive Exploration—specifically, how the agent is guided to explore the GUI without explicit instructions, the duration of exploration, or the number of exploration steps taken.

[1]. Wang J, Xu H, Jia H, et al. Mobile-agent-v2: Mobile device operation assistant with effective navigation via multi-agent collaboration[J]. arXiv preprint arXiv:2406.01014, 2024.

[2]. Agashe S, Han J, Gan S, et al. Agent s: An open agentic framework that uses computers like a human[J]. arXiv preprint arXiv:2410.08164, 2024.

[3]. Yu X, Peng B, Vajipey V, et al. Improving Autonomous AI Agents with Reflective Tree Search and Self-Learning[C]//The Thirteenth International Conference on Learning Representations. 2024.

---

> ### Author Rebuttal · Authors · 2025-07-31
>
> We sincerely thank the reviewer for the constructive and detailed feedback. We appreciate the recognition of our clear writing and effectiveness on dynamic benchmarks. Below, we address the main concerns point by point:
>
> ---
>
> > Weakness 1: The technical contribution of this paper is limited. The main contributions lie in incorporating reflectors into the GUI automation framework and leveraging historical operation information to improve operational accuracy, both of which have already been explored in prior work [1,2,3].
>
> We'd like to clarify that **reflection and exploration are broad scopes, and different works can focus on different aspects**. We declare the technical contribution of our work compared to prior efforts:
>
> Our work proposes a **real-time, hierarchical reflection mechanism** and a **pre-task proactive exploration module**, which together enhance robustness, adaptability, and generalization in dynamic mobile GUI environments. The prior works mentioned by the reviewer **differ significantly in focus and design**: [1] applies only naive single-step reflection, [2] performs post-task self-evolution relying on historical trajectories, and [3] uses reflection to assist tree search after tasks are completed. In contrast, our approach performs **online, structured, and selective reflection during execution**, and proactively builds **task-independent knowledge before execution begins**. These components are not only technically distinct but are also crucial for achieving high success rates in long-horizon, error-prone mobile scenarios, as demonstrated by our empirical results.
>
> We hope these clarifications highlight the technical novelty and significance of our contributions.  We will further emphasize our technical contribution in the revised paper.
>
> ---
>
> > Weakness 2: The paper is overly engineering-oriented. The method relies on multiple rounds of inference to achieve better performance at the expense of efficiency, which may negatively impact its practical applicability.
>
> > Weakness 3: Introducing three reflectors in the GUI automation framework inevitably increases inference time. A quantitative comparison should be provided, but such analysis is currently missing in the paper.
>
> We appreciate the reviewer’s concern regarding efficiency and practical deployment. We have conducted a detailed runtime breakdown, and our analysis shows that the overhead introduced by our hierarchical reflection framework is marginal and well justified by the performance gain:
>
> As shown in the table below. The MobileUse result corresponds to performing Action Reflection at every step, while the second uses Reflection-on-Demand (threshold = -0.01).
> - **Hierarchical reflection adds minimal latency**: With full reflection enabled, the Action, Trajectory, and Global Reflectors contribute approximately 25%, 3.5%, and 2% of the total time, respectively. When applying our Reflection-on-Demand strategy, the reflection overhead is significantly reduced to 10%, while the success rate drops by less than 1%, showing that unnecessary computation is effectively avoided without sacrificing performance.
> - **Our method achieves both efficiency and performance gains**: Under the same backbone (Qwen2.5-VL-72B), MobileUse with Reflection-on-Demand runs faster than Mobile-Agent-v2 and achieves a much higher success rate on AndroidWorld (>20%), demonstrating the effectiveness of our hierarchical reflection design.
>
> | Method | Module | Time ($s$/task) | AndroidWorld_SR (%) |
> |---|---|---|---|
> | Mobile-Agent-v2 | Total | 319 | 37.1 |
> | MobileUse | Total | 359 / 279 | 58.6 / 57.8 |
> | MobileUse | Action Reflector | 90 / 10.6  | \ |
> | MobileUse | Trajectory Reflector | 13.2 | \ |
> | MobileUse | Global Reflector | 7.7 | \ |
>
> ---
>
> > Weakness 4: As shown in the ablation results (Table 4), the model in the fourth row achieves better performance on Medium and Hard tasks, whereas Reflection-on-Demand and Proactive Exploration primarily benefit Easy tasks and can even significantly reduce performance on more difficult tasks. The paper lacks any explanation for this phenomenon. Since Easy tasks constitute the majority in the AndroidWorld benchmark, the full model appears strong overall, but these results suggest that Reflection-on-Demand and Proactive Exploration offer limited benefit for more challenging scenarios.
>
> Thank you for the insightful observation. We would like to clarify this phenomenon and share our design rationale:
> - **Reflection-on-Demand achieves a trade-off between efficiency and robustness.** We observed that in some cases, especially on simple tasks, the reflectors could produce incorrect feedback due to hallucinations, leading to unnecessary or even harmful corrections. Our Reflection-on-Demand strategy filters out low-risk steps with high action confidence, thereby avoiding incorrect reflections while preserving essential ones. This mechanism is particularly effective on easy tasks, where the agent is already less prone to mistakes. As a result, this strategy not only improves efficiency but also enhances performance by reducing interference from spurious reflections.
>
> - **Proactive Exploration benefits tasks that appear easy but are actually hard and require prior knowledge.** While some tasks are labeled as “easy” by the AndroidWorld benchmark based on surface features like short length, they are in fact difficult to complete without prior exploration—for example, when interface semantics are unfamiliar. In such cases, our Proactive Exploration module is critical, as it allows the agent to acquire general knowledge about the app beforehand. This enables successful execution where agents without such context would fail. Thus, the observed gain in “easy” tasks actually reflects improved performance on structurally hard but short tasks.
>
> We will revise the paper to provide detailed explanations of the ablation results and our future work.
>
> ---
>
> > Weakness 5: The paper does not sufficiently describe the implementation details of Proactive Exploration—specifically, how the agent is guided to explore the GUI without explicit instructions, the duration of exploration, or the number of exploration steps taken.
>
> We thank the reviewer for pointing this out and would like to provide more information:
>
> 1) *Implementation details of proactive exploration*
>
> As illustrated in Figure 1, our method adopts **a multi-agent collaboration framework to guide the exploration process**:
> (i) We guide the **Action Agent** explores unfamiliar regions of the current app environment without relying on task-specific instructions; (ii) The **Summary Agent** analyzes the environment states before and after actions, along with executed action information, to generate and update reusable knowledge; (iii) The **Judge Agent** monitors the exploration process to prevent redundant or ineffective behavior and provides corrective feedback when necessary. By combining these agents, the model can proactively collect common knowledge, which is especially useful when dealing with scenarios involving exceed the boundaries of their pre-trained capabilities.
>
> 2）*Temporal and computational cost of the exploration module*
>
> In terms of resource usage, we set the total number of exploration steps to 100 per app. The token and time cost per agent per step is as follows:
>
> Token cost per step: Action = 5,429; Summary = 8,135; Judge = 3,483.
>
> Time cost per step : Action = 3.4s; Summary = 11.5s; Judge = 4.6s.
>
> It is worth noting that exploration is performed only once per app, and the accumulated knowledge is shared across multiple downstream tasks, making the cost amortizable and practical in real deployments.
>
> We will provide these details in the revision.
>
> ---
>
> > Question 1: This paper claims that Proactive Exploration can help accumulate common knowledge, but its experimental results are not satisfactory. Proactive Exploration also incurs a time cost. If there is a domain gap between the environment explored and the experimental environment, can Proactive Exploration still be effective?
>
> We appreciate the reviewer’s concerns and would like to provide further clarification:
>
> 1. *Irreplaceability of Proactive Exploration*
>
> **Proactive Exploration plays an essential role in certain tasks, where prior exploration of the app environment is critical**. Without this preliminary exploration, the agent would likely fail to complete the task. This is particularly true for apps with unfamiliar structures or dynamic content, where interaction knowledge must be gathered beforehand. As shown in our experiments, the Proactive Exploration module provides significant improvements in these cases.
>
>
> 2. *Transferability of the exploration module*
>
> The summary agent in the exploration module ensures that the accumulated knowledge is stored in a universal text format.  To in-depth discuss the transferability of knowledge, we have taken into account scenarios in real-world applications, including minor version updates, UI changes, similar applications, different operating systems, and cross-device situations.
>
> Taking the Chrome browser as an example, which is found in both AndroidWorld and daily usage: The exploration module summarizes operations such as searching, recommendation discovery, shortcuts to popular websites, information discovery, app downloading and installation, settings, and file opening. We find that most of the summarized knowledge remains consistent across different app versions and operating systems. Since the knowledge does not involve UI interface information, it is unaffected by UI changes that do not alter interaction logic. Incompatibility issues may arise only during knowledge transfer between similar apps, but these can often be resolved through further exploration and adaptation using the experience from similar apps.
>
> We will revise the manuscript to further clarify the significance of our proactive exploration and its transferability.

---

> > ### Comment · Reviewer_1iG5 · 2025-08-07
> >
> > Thank you very much for the detailed response.  However, some concerns remain regarding the technical contributions.  The subtle differences clarified between the proposed approach and existing methods do not constitute strong evidence of a technical contribution.
> >
> > Nevertheless, some questions persist about the time overhead.  While new experimental results are not required, clarification on several points would be appreciated:
> >
> > - In weakness2's reply, on which benchmark was the time overhead statistically measured?
> >
> > - The increased reflector appears likely to increase the number of steps required to achieve success rate contribution.  Are there any existing statistical results documenting the overhead caused by this increase?
> >
> > - Concerns also exist about the baseline used in the experiments.  The test scenario for mobile-agent-v2 was reportedly conducted on real devices, which differs from the current benchmark (AndroidWorld?).  An explanation would be welcome regarding why the method was not compared with the baseline provided by AndroidWorld.
> >
> > These points are offered to help refine and clarify the research, and assist me in making my final decision.

---

> > > ### Author Response · Authors · 2025-08-07
> > >
> > > We sincerely appreciate the reviewer’s continued engagement and the opportunity to address the additional concerns raised. Below, we provide further clarification on the points raised regarding time overhead and technical contribution.
> > >
> > > ---
> > >
> > > > In weakness2's reply, on which benchmark was the time overhead statistically measured?
> > >
> > > The time overhead analysis of Mobile-Agent-v2 and MobileUse was performed on the **AndroidWorld** benchmark, as it is a widely used and representative environment for evaluating mobile GUI automation tasks.
> > >
> > > ---
> > >
> > > > The increased reflector appears likely to increase the number of steps required to achieve success rate contribution. Are there any existing statistical results documenting the overhead caused by this increase?
> > >
> > > We calculated the average number of steps per task on the AndroidWorld benchmark across various settings, as shown in the table below. We can find that:
> > > - The introduction of reflection mechanisms results in a slight increase in the number of steps per task. **This is intuitive, as error detection and correction naturally require additional steps.** However, it is essential to note that this increase in steps is accompanied by a significant improvement in the success rate, thereby demonstrating the benefits of incorporating reflection for error correction and enhanced robustness.
> > > - **The hierarchical reflection framework does not lead to a substantial increase in the number of steps compared to the single-step reflection (<0.5 steps). Moreover, the Reflection-on-Demand strategy reduces the number of steps required while improving performance**, showcasing the efficiency of our approach in balancing reflection complexity and task execution time.
> > > - Importantly, the **AndroidWorld benchmark imposes a maximum step limit for each task**, and **all settings were evaluated within this unified maximum step constraint**. Therefore, the increase in step count due to the reflection mechanism does not exceed the predefined limits. This further highlights the validity of our approach in optimizing both efficiency and task success.
> > >
> > > |Settings|#Steps / Task|SR (%)|
> > > |---|---|---|
> > > |Operator + Progressor|10.02|49.5|
> > > |+) Action Reflector|12.60|55.2|
> > > |+) Trajectory Reflector|13.01|56.1|
> > > |+) Global Reflector|13.03|58.6|
> > > |+) Reflection-on-Demand|12.61|61.6|
> > >
> > > ---
> > >
> > > > Concerns also exist about the baseline used in the experiments. The test scenario for mobile-agent-v2 was reportedly conducted on real devices, which differs from the current benchmark (AndroidWorld?). An explanation would be welcome regarding why the method was not compared with the baseline provided by AndroidWorld.
> > >
> > > We appreciate the reviewer’s concern regarding the baseline comparison and would like to clarify the following points:
> > > - **Architecture Difference**: Compared to the simple design of M3A (baseline provided by AndroidWorld), Mobile-Agent-v2 is a multi-agent framework that incorporates a naive single-step reflection module. This design is more aligned with our approach, providing a more comparable baseline for our experiments.
> > > - **Input Difference**: M3A includes the a11y tree as input, whereas MobileUse operates solely with screenshots as input, which is more generalizable across a broader range of mobile environments.
> > > - **Significant Performance Gap**: The AndroidWorld baseline exhibits relatively low performance, which further diminishes the relevance and significance of directly comparing it.
> > >
> > > Despite these differences, we have included the results from M3A, as reported in the AndroidWorld paper, in the table below. As shown, while our MobileUse framework incurs a slightly higher execution time, it significantly outperforms the baseline methods in terms of task success rate. This highlights the effectiveness of our hierarchical reflection approach.
> > >
> > > |Method|Base Model|Time ($s$/task)|SR (%)|
> > > |---|---|---|---|
> > > |Mobile-Agent-v2|Qwen2.5-VL-72B|319|37.1|
> > > |MobileUse|Qwen2.5-VL-72B|359 / 279|58.6 / 57.8|
> > > |M3A (screen + a11y tree)|GPT-4 Turbo|234|25.4|
> > > |M3A (a11y tree)|GPT-4 Turbo|150|30.6|
> > >
> > > ---
> > >
> > > Finally, we would like to emphasize our technical contribution:
> > >
> > > Our hierarchical reflection architecture goes beyond the naive single-step reflection approach by incorporating the **Reflection-on-Demand mechanism** at the action level to achieve **a dual benefit of efficiency and performance**, a trajectory-level reflection to **ensure consistency in goal attainment**, and a global-level reflection to **offer comprehensive task oversight**. These hierarchical reflection mechanisms **work in synergy, collectively enhancing the robustness and stability of task execution**.
> > >
> > > Additionally, our proactive exploration architecture surpasses the passive, task-driven exploration approaches, allowing the agent to interact with the environment **proactively**. This facilitates the acquisition of **a broader range of common knowledge**, thereby improving the agent's robustness when dealing with unfamiliar environments.

---

> > > > ### Comment · Reviewer_1iG5 · 2025-08-08
> > > >
> > > > Thank you for the detailed response and additional experimental results. Although the proposed method introduces additional computational overhead, it achieves a notable improvement in success rate.
> > > >
> > > > However, I have one reservation - the method seems to be designed specifically for GUI automation tasks, rather than solely for improving the accuracy on the AndroidWorld benchmark. After all, AndroidWorld only contains around 100 test cases. The author's mention of the "AndroidWorld benchmark imposing a maximum step limit for each task" suggests that the method may not be readily applicable to real-world scenarios, which raises concerns about its generalization capability.
> > > >
> > > > Based on the current results, while the technical contribution, method efficiency, and generalization ability could be improved, the performance on the AndroidWorld benchmark is indeed impressive. After careful consideration of the rebuttal and the feedback from other reviewers, I have decided to increase my score to 4.

---

> > > > > ### Author Response · Authors · 2025-08-08
> > > > >
> > > > > We sincerely thank the reviewer for the thoughtful follow-up and for recognizing the significant performance improvements achieved by our method. We truly appreciate your decision to raise the score based on our responses and the broader discussion.
> > > > >
> > > > > We would like to clarify that our framework is not explicitly designed for the AndroidWorld benchmark, but rather for real-world GUI automation scenarios. AndroidWorld serves as a standardized and widely-adopted benchmark to validate system performance in a reproducible manner. Still, our ultimate goal is to deploy the proposed framework in real mobile environments to support real users in completing complex tasks on their devices. **In fact, we are currently working on integrating our system into mobile OS for live deployment.**
> > > > >
> > > > > In this context, our hierarchical reflection framework offers a compelling balance between performance and computational cost. While it introduces a marginal increase in inference time, it yields substantial gains in task success rate. **We believe this trade-off is highly favorable for real-world applications, where the cost of task failure—both in terms of user experience and the need to repeat task execution—can be significantly higher than the slight computational overhead incurred by reflection.**
> > > > >
> > > > > Once again, we sincerely thank the reviewer for the constructive feedback and for acknowledging the strengths of our work.

---

> ### Author Response · Authors · 2025-08-06
>
> Dear Reviewer,
>
> I hope this message finds you well. As the discussion period is nearing its end with less than three days remaining, I wanted to ensure we have addressed all your concerns satisfactorily. If there are any additional points or feedback you'd like us to consider, please let us know. Your insights are invaluable to us, and we're eager to address any remaining issues to improve our work.
>
> Thank you for your time and effort in reviewing our paper.

---

### Official Review · Reviewer_8sCX · 2025-07-02

**Clarity:** 3
**Significance:** 3
**Originality:** 3
**Rating:** 4
**Confidence:** 4

**Summary:**

The paper presents MobileUse, a GUI agent designed for automating complex tasks on mobile devices. It introduces a hierarchical reflection architecture that enables self-monitoring and error correction during task execution, enhancing task success rates. Additionally, the proactive exploration module helps the agent accumulate knowledge in unfamiliar environments, addressing cold-start issues. The empirical results demonstrate state-of-the-art performance on benchmark tests, and the authors provide an out-of-the-box toolkit for practical applications.

**Questions:**

See Weaknesses.

**Ethical Concerns:**

["NO or VERY MINOR ethics concerns only"]

**Final Justification:**

After reading the response and the other reviewer's reviews, I appreciate the efforts and additional experiments of the authors, most of my concerns have been addressed and I have no doubt on the experiment results while also recognizing it mainly improves upon existing multi-agent framework on reflection part, therefore I maintain my positive score.

**Limitations:**

Yes

**Quality:**

3

**Strengths And Weaknesses:**

Strengths:
1. Novel Approach: The focus on reflection rather than traditional planning is innovative, setting MobileUse apart from existing work that primarily emphasizes planning strategies.
2. Unique Hierarchical Reflection Design: The hierarchical reflection mechanism is a fresh concept, addressing potential errors from single-dimension reflection by incorporating multiple levels of reflection granularity.
3. Significant Results: MobileUse achieves state-of-the-art performance with a 62.9% success rate on the Android World benchmark, underscoring the effectiveness of the proposed method.

Weaknesses:
1. Impact of Proactive Exploration: For a fair comparison, the effects of the exploration phase should be isolated, as other methods do not involve prior familiarization with the environment. Without the exploration phase, the combination of the progressor, operator, and reflectors achieves a success rate of 55.2%, which mirrors existing frameworks without novel contributions [1]. Only the global reflector shows a substantial improvement with an additional 2.5%.
2. Sensitivity of Reflection-on-Demand: While the success rate increase is notable (+3%), the Reflection-on-Demand strategy seems quite trick-oriented. Ablation studies indicate that the entire system is highly sensitive to the chosen threshold, which requires manual tuning on benchmarks. Additionally, this threshold may not be applicable across different models, posing limitations on its generalizability. Moreover, this approach is suitable mainly for open-source models, restricting its applicability in certain scenarios.
3. Sensitivity to Model Size: MobileUse's performance is heavily dependent on the model size. It would be beneficial if the authors could provide an analysis of the capabilities lacking in smaller models, which contribute to the drop in performance.
4. Typographical Errors: Upon reviewing the manuscript, I found a few typographical errors:
  ○ In Section 3.1, "pre-compute" should be hyphenated as "pre-compute."
  ○ In the same section, the word "superscript" appears twice and could potentially be simplified or clarified.
  ○ The phrase "reflection-on-demand strategy" could benefit from consistent capitalization throughout the document.
[1]: Mobile-Agent-v2: Mobile Device Operation Assistant with Effective Navigation via Multi-Agent Collaboration

---

> ### Author Rebuttal · Authors · 2025-07-31
>
> Thank you for your detailed feedback and constructive comments on our manuscript. We appreciate your recognition of the novel aspects of MobileUse and the significant results presented. We would like to address your concerns and provide additional clarifications as follows:
>
> ---
>
> > Weakness 1: Impact of Proactive Exploration: For a fair comparison, the effects of the exploration phase should be isolated, as other methods do not involve prior familiarization with the environment. Without the exploration phase, the combination of the progressor, operator, and reflectors achieves a success rate of 55.2%, which mirrors existing frameworks without novel contributions [1]. Only the global reflector shows a substantial improvement with an additional 2.5%.
>
> We sincerely thank the reviewer for raising this important point. To clarify, Table 4 in our paper provides a detailed breakdown of the performance with and without the proactive exploration phase. **The success rate without proactive exploration is 61.6%, demonstrating a 6.4% absolute improvement** compared to the setting that only includes the progressor, operator, and single-step reflection. This clearly shows that our hierarchical reflection architecture contributes significantly to performance even in the absence of proactive exploration.
>
> We would also like to emphasize the complementary roles of each reflection component:
> - Reflection-on-Demand and Action Reflection: The Reflection-on-Demand mechanism significantly improves efficiency by triggering reflection only when execution confidence drops, thereby reducing unnecessary computations and avoiding overcorrection of correct actions, achieving performance improvement while reducing the overhead. **Our designed Action Reflection combines the Reflection-on-Demand mechanism and perception module, which is significantly different from the traditional step-level reflection and has achieved a huge performance improvement**.
> - Trajectory Reflection: This module helps identify accumulated, latent errors and **ensures consistency throughout multi-step task execution**. This is especially important in real-world mobile scenarios, where GUI agents frequently fall into loops or repeated actions due to accumulated mistakes.
> - Global Reflection: This module **addresses hallucination and semantic drift** by enabling the agent to reassess its state from a global perspective. As reported, it alone contributes a notable +2.5% success rate gain, illustrating its impact.
>
> Collectively, these reflection modules work synergistically to enhance MobileUse's robustness and fault tolerance, which we believe **constitutes a meaningful and novel contribution beyond existing frameworks**—even before considering proactive exploration.
>
> ---
>
> > Weakness 2: Sensitivity of Reflection-on-Demand: While the success rate increase is notable (+3%), the Reflection-on-Demand strategy seems quite trick-oriented. Ablation studies indicate that the entire system is highly sensitive to the chosen threshold, which requires manual tuning on benchmarks. Additionally, this threshold may not be applicable across different models, posing limitations on its generalizability. Moreover, this approach is suitable mainly for open-source models, restricting its applicability in certain scenarios.
>
> We appreciate the reviewer’s comments regarding the Reflection-on-Demand strategy and its potential sensitivity to the threshold value.
>
> First, we would like to clarify that the range of thresholds shown in Figure 5 was deliberately selected to cover a wide interval so as to clearly illustrate the performance trend. In practice, the threshold offers ample flexibility: for example, when the threshold is lowered to -0.01, over 90% of reflections are filtered out, yet the overall success rate remains largely stable. This demonstrates that **our system is robust to threshold variation and not overly sensitive to specific tuning**. The choice of threshold can therefore be adjusted based on available computational resources or latency requirements, making it a controllable and practical design.
>
> Moreover, we respectfully disagree with the notion that the Reflection-on-Demand mechanism is “trick-oriented.” In fact, **it is built on a meaningful insight: Reflection, while useful, can introduce errors or inefficiencies when applied excessively**. Our strategy leverages the model’s internal confidence to selectively suppress reflection when the model is already confident, and retain reflection when the model is uncertain. This allows us to achieve a dual benefit in both performance and efficiency, by avoiding misleading corrections and focusing computational effort where it is most needed.
>
> In terms of generalizability, the threshold is derived from token-level log probabilities, which are **inherent to the language model itself and independent of specific tasks or environments**. Therefore, while the optimal value may vary slightly depending on the model used, it is generally model-dependent but task-agnostic, and requires only once-per-model calibration in practical applications.
>
> Lastly, we would like to point out that many closed-source models (e.g., GPT-4, GPT-4o) also expose token-level logprobs or similar confidence signals through their APIs. Thus, **our approach remains compatible with a wide range of both open- and closed-source models**, and is not fundamentally restricted in scope.
>
> We will revise the manuscript to clarify these theoretical motivations, practical considerations, and compatibility aspects in more detail.
>
> ---
>
> > Weakness 3: Sensitivity to Model Size: MobileUse's performance is heavily dependent on the model size. It would be beneficial if the authors could provide an analysis of the capabilities lacking in smaller models, which contribute to the drop in performance.
>
> We thank the reviewer for pointing out the performance dependence on model size. In Appendix D.2, we already provided empirical results using smaller vision-language models—Qwen-2.5-VL-7B and Qwen-2.5-VL-32B—which achieved 21.6% and 44.4% success rates, respectively. We also included a specific analysis of the performance degradation from different aspects. Here, we elaborate further on the specific capability gaps we observed in smaller models that contribute to this drop:
> - **Instruction-Following Ability**: We found that Qwen-2.5-VL-7B often fails to follow structured instructions and struggles to produce actions in the correct format (e.g., missing keys in JSON actions), which is a critical requirement for acting as an agent. This limits its usability without significant post-processing or constrained decoding.
> - **Grounding Ability**: Both the 7B and 32B variants frequently exhibit grounding errors, where the model misidentifies UI elements or fails to map visual features to corresponding semantic entities. Since grounding is arguably the most essential skill for any GUI agent, this significantly impacts task success.
> - **Planning and Reasoning Ability**: Smaller models also lack strong task-level reasoning capabilities. They often fail to understand the global objective of the task or break it down into coherent subgoals, resulting in inefficient or incorrect action sequences. Moreover, their limited reasoning ability undermines the effectiveness of our reflection modules, which rely on the model’s capacity to infer causes of failure and plan recovery.
>
> While these limitations are inherent in current smaller models, we see promising directions for future work. Specifically, we plan to explore instruction tuning and reinforcement learning-based post-training to enhance small models’ alignment with GUI agent demands. We believe that with task-specific fine-tuning, even lightweight models can become viable components within the MobileUse framework, enabling broader applicability in resource-constrained environments.
>
> ---
>
> > Weakness 4: Typographical Errors: Upon reviewing the manuscript, I found a few typographical errors: ○ In Section 3.1, "pre-compute" should be hyphenated as "pre-compute." ○ In the same section, the word "superscript" appears twice and could potentially be simplified or clarified. ○ The phrase "reflection-on-demand strategy" could benefit from consistent capitalization throughout the document.
>
> We sincerely thank the reviewer for the careful reading and detailed suggestions regarding writing and formatting. We will address all noted issues in the revision. We appreciate your attention to these details, which will help us improve the overall quality and readability of the paper.

---

> ### Author Response · Authors · 2025-08-06
>
> Dear Reviewer,
>
> I hope this message finds you well. As the discussion period is nearing its end with less than three days remaining, I wanted to ensure we have addressed all your concerns satisfactorily. If there are any additional points or feedback you'd like us to consider, please let us know. Your insights are invaluable to us, and we're eager to address any remaining issues to improve our work.
>
> Thank you for your time and effort in reviewing our paper.

---

> ### Author Response · Authors · 2025-08-08
>
> Dear Reviewer,
>
> We sincerely appreciate your positive recognition of the novelty in our approach and design, as well as your acknowledgement of the strong empirical performance of MobileUse.
>
> Regarding the concerns you raised, we have provided detailed clarifications in our rebuttal, including:
> - Isolating the contribution of proactive exploration and showing that the hierarchical reflection design itself yields substantial gains.
> - Explaining the motivation, robustness, and broad applicability of the Reflection-on-Demand strategy.
> - Analyzing the capability gaps in smaller models to account for the observed performance drop.
> - Addressing all noted typographical and formatting issues.
>
> We hope these explanations have addressed your questions and provided more clarity on our design choices and contributions. If there are any remaining doubts or points you would like us to elaborate on further, we would be glad to discuss them in detail. We look forward to your feedback and any further questions you may have.
>
> Thank you again for your valuable time and constructive comments.

---

### Official Review · Reviewer_c8Yj · 2025-07-03

**Clarity:** 4
**Significance:** 4
**Originality:** 4
**Rating:** 4
**Confidence:** 3

**Summary:**

The paper introduces **MobileUse**, a hierarchical reflection-based GUI agent designed for robust and adaptive task execution on mobile devices using multimodal large language models (MLLMs). It tackles three core challenges in mobile automation: long-horizon task execution, error recovery, and cold-start scenarios in unfamiliar environments. The framework consists of two key components: a Hierarchical Reflection Architecture with Action, Trajectory, and Global Reflectors that self-monitor across temporal scales via a "Reflection-on-Demand" mechanism, and a Proactive Exploration Module that enables learning from open-ended interactions without predefined tasks. MobileUse achieves state-of-the-art success rates of 62.9% on AndroidWorld and 44.2% on AndroidLab, and is accompanied by an open-source toolkit for real-world deployment.

**Questions:**

In addition to the weaknesses noted above, I have a few minor questions for clarification:

1. Cross-App Knowledge Transfer: To what extent can the knowledge acquired during proactive exploration (e.g., identifying a red checkbox as indicating high priority) generalize across different app versions or to similar apps?

2. Edge Deployment Feasibility: Considering MobileUse’s reliance on large-scale language models, is it doable for distilling or optimizing the agent to enable deployment on resource-constrained devices (e.g., mobile or edge hardware)?

3. Privacy Concerns: Since MobileUse processes visual data directly from user screens—which may include sensitive information such as messages, financial details, or personal identifiers—what mechanisms are in place to ensure user privacy and data security?

**Ethical Concerns:**

["NO or VERY MINOR ethics concerns only"]

**Final Justification:**

I have read through the authors’ rebuttal and other reviewers’ comments. While I agree that the method and novelty could be improved (based on Reviewer 1iG5, it currently feels more like an integration of existing methods). I remain concerned about the generalization ability across different mobile systems, devices, layouts, etc., as well as privacy issues (specifically, whether all processing can be done locally on-device). Overall, the paper is good, and the authors’ responses have addressed most of my concerns. I will keep my current rating.

**Limitations:**

Please refer to the Weakness and Limitation sections.

**Paper Formatting Concerns:**

The reference format does not follow NeurIPS.

**Quality:**

3

**Strengths And Weaknesses:**

## **Strengths**
1. Effective Architecture
The paper introduces MobileUse, a well-structured hierarchical reflection mechanism that catches errors across different execution levels, offering clear advantages over flat, monolithic models. The Reflection-on-Demand strategy further enhances robustness by triggering reflection only when needed, striking a smart balance between efficiency and error recovery.

2. Generalization and Adaptability
The proactive exploration phase helps the agent with prior knowledge of apps, enabling effective adaptation without relying on labeled data.

3. Real-World Validation
With the release of the MobileUse Toolkit, the system demonstrates strong practical relevance by supporting real-device automation. The paper also includes thorough evaluation—ablation studies, confusion matrices, and failure analysis.

## **Weaknesses**
1. Scalability and Accessibility Concerns
MobileUse depends on large models (up to 72B parameters), with smaller variants (e.g., 7B, 32B) exhibiting significant performance degradation. This will make the model rely on internet access and limit the approach’s accessibility, practicality, and potential for deployment on resource-constrained edge devices (i.e., phones).

2. Incomplete Evaluation of Generalization and Efficiency
The paper lacks an analysis of MobileUse's ability to generalize to entirely novel or unseen apps. It also ignores any discussion of computational cost or latency introduced by the reflection modules—critical factors for real-time use on mobile devices.

3. Limited Adaptation: The system does not explicitly handle tasks requiring human validation (e.g., biometric login, payment confirmation), which are common in real-world apps. Additionally, while proactive exploration helps, the paper lacks in-depth discussion on how MobileUse handles app version changes, OS-specific UI variations, and different devices/systems.

---

> ### Author Rebuttal · Authors · 2025-07-31
>
> Thank you for the thoughtful review and for highlighting the strengths of our work—especially the hierarchical reflection design, Reflection-on-Demand strategy, proactive exploration method, and real-world relevance. We address your concerns point-by-point below.
>
> ---
>
> > Weakness 1: Scalability and Accessibility Concerns MobileUse depends on large models (up to 72B parameters), with smaller variants (e.g., 7B, 32B) exhibiting significant performance degradation. This will make the model rely on internet access and limit the approach’s accessibility, practicality, and potential for deployment on resource-constrained edge devices (i.e., phones).
>
> Thank you for raising this important point. We agree that the reliance on large models presents challenges for edge deployment and have discussed it in Section 5. Here we'd like to make a further clarification:
> - **Limitations of Small Models**: As analyzed in Appendix D.2, smaller models often struggle with the complex reasoning and grounding capabilities. Our experiments show that while smaller models can execute simpler tasks, their performance degrades significantly on tasks involving fine-grained perception and complex reasoning.
> - **Industry Practice**: Currently, both academic and industrial GUI agents [1, 2] rely on cloud-based large models due to these performance gaps. **MobileUse aligns with this trend to explore the upper-bound of automation capabilities.**
> - **Ongoing Optimization**: We are actively developing a training pipeline to fine-tune smaller models using GUI trajectories. Early results are promising, and we plan to report them in future work.
>
> ---
>
>
> > Weakness 2.1: Incomplete Evaluation of Generalization and Efficiency The paper lacks an analysis of MobileUse's ability to generalize to entirely novel or unseen apps.
>
> Thank you for raising this important point. We would like to clarify that before introducing the proactive exploration module, MobileUse was evaluated in a **cold-start setting, without being deliberately trained or adapted to unseen apps** (e.g., 61.6% on AndroidWorld), demonstrating the core architecture's strong generalization ability.
>
> Furthermore, we observed in certain scenarios that even large MLLMs can struggle without prior exposure to common UI semantics. To address these shortcomings, we designed the **Proactive Exploration module to to improve robustness and adaptation to unseen app variants**.
>
> ---
>
> > Weakness 2.2: It also ignores any discussion of computational cost or latency introduced by the reflection modules—critical factors for real-time use on mobile devices.
>
> We thank the reviewer for highlighting this important point. We have conducted a detailed runtime breakdown, and our analysis shows that **the overhead introduced by our hierarchical reflection framework is marginal and well justified by the performance gain**.
>
> As shown in the table below. The first MobileUse result corresponds to performing Action Reflection at every step, while the second uses Reflection-on-Demand (threshold = -0.01). We can find that:
> - **Hierarchical reflection adds minimal latency**: With full reflection enabled, the Action, Trajectory, and Global Reflectors contribute approximately 25%, 3.5%, and 2% of the total time, respectively. When applying our Reflection-on-Demand strategy, the reflection overhead is significantly reduced to 10%, while the success rate drops by less than 1%, showing that unnecessary computation is effectively avoided without sacrificing performance.
> - **Our method achieves both efficiency and performance gains**: Under the same backbone (Qwen2.5-VL-72B), MobileUse with Reflection-on-Demand runs faster than Mobile-Agent-v2 and achieves a much higher success rate on AndroidWorld (>20%), demonstrating the effectiveness of our hierarchical reflection design.
>
> | Method | Module | Time ($s$/task) | AndroidWorld_SR (%) |
> |---|---|---|---|
> | Mobile-Agent-v2 | Total | 319 | 37.1 |
> | MobileUse | Total | 359 / 279 | 58.6 / 57.8 |
> | MobileUse | Action Reflector | 90 / 10.6  | - |
> | MobileUse | Trajectory Reflector | 13.2 | - |
> | MobileUse | Global Reflector | 7.7 | - |
>
> ---
>
> > Weakness 3.1: Limited Adaptation: The system does not explicitly handle tasks requiring human validation (e.g., biometric login, payment confirmation), which are common in real-world apps.
>
> We appreciate the reviewer’s observation regarding real-world tasks that require explicit human involvement. We would like to offer several clarifications and forward-looking directions:
> - Current Scope: As GUI agents remain an emerging research direction, current efforts—including ours—**focus primarily on increasing robustness and success rates in autonomous mobile operation tasks** (e.g., navigation, information seeking, settings adjustment). These tasks represent a large fraction of real-world mobile usage and do not typically involve human-in-the-loop security confirmations.
> - In practical deployment, we envision two complementary strategies for handling such tasks within the MobileUse framework: (i) **Pausing and Deferring**: We can extend the operator’s action space to include a call-user operation, allowing MobileUse to recognize gated states and defer task execution until user validation is complete. (ii) **Specialized Scene Detectors**: We can integrate lightweight classifiers or on-device detectors to identify scenarios requiring human validation.
> - **Framework Flexibility**: Importantly, MobileUse’s modular design—with separable operator, reflection, and progressor modules—allows for seamless incorporation of either strategy without architectural modification.
>
> ---
>
> > Weakness 3.2: Additionally, while proactive exploration helps, the paper lacks in-depth discussion on how MobileUse handles app version changes, OS-specific UI variations, and different devices/systems.
>
> > Question 1: Cross-App Knowledge Transfer: To what extent can the knowledge acquired during proactive exploration (e.g., identifying a red checkbox as indicating high priority) generalize across different app versions or to similar apps
>
> We thank the reviewer for this insightful question.
>
> **The knowledge accumulated during Proactive Exploration is stored in a universal text format, allowing it to generalize across environments.** Taking the Chrome browser as an example, which is found in both AndroidWorld and daily usage: The exploration module summarizes operations such as searching, recommendation discovery, shortcuts to popular websites, information discovery, app downloading and installation, settings, and file opening. We find that most of the summarized knowledge remains consistent across different app versions and operating systems. Since the knowledge does not involve UI interface information, it is unaffected by UI changes that do not alter interaction logic.
>
> While there may be some incompatibility issues when transferring knowledge across very different apps, these challenges can typically be addressed with further exploration, adapting the learned experiences from similar apps.
>
> We will revise the manuscript to further clarify the significance of our proactive exploration and its transferability.
>
> ---
>
> > Question 2: Edge Deployment Feasibility: Considering MobileUse’s reliance on large-scale language models, is it doable for distilling or optimizing the agent to enable deployment on resource-constrained devices (e.g., mobile or edge hardware)?
>
> Thank you for this valuable question. One of the strengths of MobileUse is its **modular and model-agnostic design, allowing the architecture to be adapted to smaller, more efficient language models for edge deployment**.
>
> To this end, we are actively exploring model adaptation strategies to improve performance under limited resources. Specifically, we are constructing targeted trajectories and reflection data to fine-tune smaller MLLMs. Early exploration has shown promising results while offering significantly lower memory and latency costs. We see this as a promising direction and plan to present a more comprehensive evaluation in future work. We will also add a brief discussion on this in the revised version of the paper to clarify MobileUse’s potential for lightweight deployment.
>
> ---
>
> > Question 3: Privacy Concerns: Since MobileUse processes visual data directly from user screens—which may include sensitive information such as messages, financial details, or personal identifiers—what mechanisms are in place to ensure user privacy and data security?
>
> We take privacy very seriously. We'd like to clarify that MobileUse **does not store any user data in the cloud**. For development and testing, MobileUse supports running entirely within **Android emulators**, where no real user data is ever exposed. Besides, MobileUse is **compatible with both local and cloud-based deployment** of the underlying language model. Whenever feasible, we recommend local deployment to keep all computation on-device and avoid transmitting screen content to external servers.
>
> As part of future work, we plan to further explore lightweight local models (as mentioned earlier), as well as integrate data encryption, secure computation, or even federated inference techniques to enhance privacy guarantees in real-world settings.
>
> We emphasize that the code is **open-sourced**, and we welcome community review and improvement to further strengthen privacy safeguards. We will include a short section on privacy in the final version of the paper to make these points more explicit.
>
> ---
>
> > Paper Formatting Concerns: The reference format does not follow NeurIPS.
>
> We sincerely thank the reviewer for the careful reading and concerns regarding formatting. We will correct all formatting issues to align with NeurIPS standards in the revision.
>
> ---
>
> [1] Ui-tars: Pioneering automated gui interaction with native agents.
>
> [2] Agent s: An open agentic framework that uses computers like a human.

---

> ### Author Response · Authors · 2025-08-06
>
> Dear Reviewer,
>
> I hope this message finds you well. As the discussion period is nearing its end with less than three days remaining, I wanted to ensure we have addressed all your concerns satisfactorily. If there are any additional points or feedback you'd like us to consider, please let us know. Your insights are invaluable to us, and we're eager to address any remaining issues to improve our work.
>
> Thank you for your time and effort in reviewing our paper.

---

### Author Response · Authors · 2025-08-08

We sincerely thank all reviewers for their detailed, insightful, and constructive feedback. We are encouraged by the following reviewers' perceptions:
- **Novel focus on reflection and unique hierarchical reflection architecture** (c8Yj, 8sCX, Fa8z).
- **Significant performance improvements** (c8Yj, 8sCX, 1iG5, Fa8z).
- **Comprehensive ablation studies and analysis** (c8Yj, Fa8z).
- **Clear and well-structured writing** (c8Yj, 1iG5).

We appreciate the reviewers’ valuable suggestions and questions, which helped refine our work and guide future research. We have addressed all concerns and questions in individual responses. Below we highlight some key points.

---

> Efficiency and overhead of MobileUse.

1) *Hierarchical Reflection*

The overhead introduced by hierarchical reflection is marginal and well justified by the performance gain: The Action, Trajectory, and Global Reflectors contribute approximately 25%, 3.5%, and 2% of total runtime, respectively. Using Reflection-on-Demand reduces reflection overhead to about 10% with a less than 1% success rate drop. Under the same backbone, MobileUse runs faster than Mobile-Agent-v2 while achieving a success rate more than 20% higher on AndroidWorld.

|Method|Module|Time ($s$/task)|AndroidWorld_SR (%)|
|---|---|---|---|
|Mobile-Agent-v2|Total|319|37.1|
|MobileUse|Total| 359 / 279 | 58.6 / 57.8 |
|MobileUse| Action Reflector | 90 / 10.6  | - |
|MobileUse| Trajectory Reflector | 13.2 | - |
|MobileUse| Global Reflector | 7.7 | - |

2) *Proactive Exploration*

We set the total number of exploration steps to 100 per app. Proactive exploration is performed only once per app, and the accumulated knowledge is shared across multiple downstream tasks, making the cost amortizable and practical in real deployments. The token and time cost per agent per step is as follows:

Token cost per step: Action = 5,429; Summary = 8,135; Judge = 3,483.
Time cost per step : Action = 3.4s; Summary = 11.5s; Judge = 4.6s.

---

> Technical contribution.

Our hierarchical reflection architecture goes beyond the naive single-step reflection approach by incorporating the **Reflection-on-Demand mechanism** at the action level to achieve **a dual benefit of efficiency and performance**, a trajectory-level reflection to **ensure consistency in goal attainment**, and a global-level reflection to **offer comprehensive task oversight**. These hierarchical reflection mechanisms **work in synergy, collectively enhancing the robustness and stability of task execution**.

Our proactive exploration architecture **surpasses the passive, task-driven exploration approaches, allowing the agent to interact with the environment proactively**. This facilitates the acquisition of a broader range of common knowledge, thereby improving the agent's robustness when dealing with unfamiliar environments.

Both modules target the same overarching challenge: improving the robustness and adaptability of GUI agents in real-world mobile tasks.

---

> Sensitivity of Reflection-on-Demand and model size.

**Reflection-on-Demand is highly stable**: Even when reducing over 85% of reflections, performance drops by less than 1%. The threshold depends only on model-internal confidence scores, making it model-dependent but task-agnostic, and requiring only one-time calibration per model. This design works across diverse tasks and datasets, and is compatible with both open- and closed-source models that expose confidence signals, ensuring broad applicability without being overly sensitive to fine-tuning.

**Limited capabilities of small models**: Small models exhibit notable deficiencies in instruction-following, UI grounding, and task-level reasoning, leading to lower success rates on complex, long-horizon tasks. In practice, both academia and industry commonly employ large models for GUI agents, as they enable rapid framework validation, optimization, and higher real-world usability. Consistent with this trend, our current implementation uses large models to explore the upper bound of performance, while we are actively developing fine-tuning pipelines to improve smaller models for broader deployment.

---

> Lake of baselines.

We have already included comparisons with several recent and strong End-to-End agents and multi-agent frameworks. Below, we added the results for CogAgent and Mobile-Agent-v2 on the AndroidWorld benchmark. Although CogAgent is fine-tuned for GUI automation tasks, it performs poorly due to its insufficient grounding and reasoning ability. When comparing with Mobile-Agent-v2, we control a consistent backbone model. Our MobileUse achieves significant performance improvement compared to the Mobile-Agent-v2 framework, attributed to the hierarchical reflection architecture for error recovery and proactive exploration for common knowledge accumulation.
|Method|Model|SR|
|-|-|-|
|CogAgent|CogAgent-9B-20241220|9.0|
|Mobile-Agent-v2|Qwen2.5-VL-72B|37.1|
|MobileUse|Qwen2.5-VL-72B|62.9|

---

### Note · Authors · 2025-08-12

We sincerely thank all reviewers for their detailed feedback and valuable discussions throughout the rebuttal process. In our official comment to all reviewers and ACs, we summarized the positive remarks and responses to key concerns. Below, we provide a further summary of the rebuttal process and the contributions of our work.

---

**Review Summary**

We are encouraged by the positive feedback from Reviewer c8Yj and 8sCX and appreciate the productive discussions with Reviewer 1iG5 and Fa8z, which led to their supportive responses:
- This paper addresses an important issue in the GUI Agent scenario: how to enable models to possess reflective capabilities. (Fa8z)
- Unique Hierarchical Reflection Design: ... incorporating multiple levels of reflection granularity. (8sCX)
- Real-World Validation With the release of the MobileUse Toolkit, ... thorough evaluation—ablation studies, confusion matrices, and failure analysis. (c8Yj)
- This paper achieves strong performance on both ... datasets. (1iG5)

---

**Additional Experiments for Rebuttal**

In response to reviewers' concerns, we conducted additional experiments and provided thorough clarifications:
- More baselines and aligned base models: We add the results of CogAgent and Mobile-Agent-v2(with the same base model Qwen2.5VL-72B) on AndroidWorld. MobileUse significantly outperforms both, achieving a 62.9% success rate, while CogAgent scores 9.0% and Mobile-Agent-v2 reaches 37.1%.
- Latency analysis: The overhead of hierarchical reflection is minimal and justified by performance gains. With Reflection-on-Demand, the inference time only increases 10%, surpassing Mobile-Agent-v2 in both efficiency and performance. Reflection-on-Demand can reduce more than 85% reflections with less than 1% performance loss.

---

**Significant Contribution**

We firmly believe that our work makes a significant contribution to mobile GUI research:
- MobileUse is an automated, efficient, and generalized framework, representing a significant step toward applying GUI agents in real-world scenarios.
- Hierarchical reflection enhances the agent's error detection and recovery capabilities at multiple levels during task execution, achieving a dual benefit of efficiency and performance with a reflection-on-demand mechanism.
- Proactive exploration enables task-agnostic autonomous exploration of the environment, further improving the robustness and adaptability of GUI agents in real-world mobile tasks.

---

### Decision · Program_Chairs · 2025-09-17

**Decision:**

Accept (poster)

**Comment:**

This paper introduces MobileUse, a hierarchical reflection-driven GUI agent designed for robust autonomous operation on mobile devices. The proposed system addresses three core challenges in mobile automation: long-horizon task execution, error recovery, and cold-start scenarios in unfamiliar environments. The key contributions include a hierarchical reflection architecture with multi-level self-monitoring and a proactive exploration module, which together enable improved resilience, adaptability, and state-of-the-art performance on AndroidWorld and AndroidLab benchmarks. An open-source toolkit is also released, demonstrating the practicality of the approach for real-world deployment.

In general, the paper is well written, clearly motivated, and introduces a novel hierarchical reflection framework that goes beyond existing single-step reflection methods. Reviewers appreciated the thorough experimental evaluation, the significant performance improvements over prior baselines, and the release of a toolkit that enhances the paper’s real-world relevance. The reflection-on-demand strategy and proactive exploration module were noted as innovative aspects that contribute to both efficiency and adaptability.

However, as pointed out by the reviewers, there are some concerns about the work. Specifically, the technical novelty may appear incremental relative to prior GUI automation frameworks, and the reliance on very large models raises questions about scalability and accessibility in resource-constrained settings. Reviewers also highlighted the need for deeper analysis of generalization to unseen apps, the computational overhead of the reflection modules, and the sensitivity of the reflection-on-demand mechanism to hyperparameters. Despite these concerns, the authors provided convincing rebuttals with additional experiments and clarifications, which alleviate many of the reviewers’ doubts.

Overall, the paper presents a strong empirical contribution with a novel hierarchical reflection design, and it achieves clear improvements over existing baselines while demonstrating practical applicability